

**The distribution of methylated sulfur compounds, DMS and DMSP, in**
**Canadian Subarctic and Arctic marine waters during summer, 2015**
Tereza Jarníková [(1)], John Dacey[(2)], Martine Lizotte[(3)], Maurice Levasseur [(3)]
and Philippe Tortell[(1,4,5)]
(1) Department of Earth, Ocean and Atmospheric Sciences, University of British
Columbia, 2022 Main Mall, Vancouver BC, Canada V6T 1Z3
(2) Woods Hole Oceanographic Institution, Woods Hole, MA 02543.
(3) Université Laval, Department of Biology (Québec-Océan), Québec City, Québec,
Canada.
(4) Department of Botany, University of British Columbia, 6270 University Blvd.,
Vancouver BC, Canada V6T 1Z4
(5) Peter Wall Institute for Advanced Studies, University of British Columbia, 6330
Crescent Blvd., Vancouver BC, Canada V6T 1Z4
**Correspondence to: Tereza Jarníková (tjarniko@eoas.ubc.ca)**





**Abstract**
**We present seawater concentrations of dimethylsulfide (DMS), and**
**dimethylsulfoniopropionate (DMSP) measured across a transect from the Labrador**
**Sea to the Canadian Arctic Archipelago, during summer 2015. Using an automated**
**ship-board gas chromatography system, and a membrane-inlet mass spectrometer,**
**we measured a range of DMS (~1 nM to 18nM) and DMSP concentrations (~1 nM to**
**150 nM) that was consistent with previous observations in the Arctic Ocean. The**
**highest DMS and DMSP concentrations occurred in a localized region of Baffin Bay,**
**where surface waters were characterized by high chlorophyll *a* (chl *a*) fluorescence,**
**indicative of elevated phytoplankton biomass. Across the full sampling transect,**
**there were only weak relationships between DMS/P, chl *a* fluorescence and other**
**measured variables, including positive relationships between DMSP:chl *a* ratios and**
**several taxonomic marker pigments, and elevated DMS/P concentrations in partially**
**ice-covered areas. Our high spatial resolution measurements allowed us to examine**
**DMS variability over small scales (<1 km), and document strong DMS**
**concentration gradients across surface hydrographic frontal features. The new**
**observations presented in this study constitute a significant contribution to the**
**existing Arctic DMS/P dataset, and provide a baseline for future measurements in**
**the region.**



### 1. Introduction

The trace gas dimethylsulfide (DMS), a degradation product of the algal
metabolite dimethylsulfoniopropionate (DMSP), is the largest natural source of sulfur to
the atmosphere, accounting for over 90% of global biogenic sulfur emissions  (Simó,
2001). In the atmosphere, DMS is rapidly oxidized to sulfate aerosols that act as cloud
condensation nuclei (CCN), backscattering incoming radiation, increasing the albedo of
low-altitude clouds and potentially cooling the Earth (Charlson et al., 1987). The seminal
CLAW hypothesis proposed by Charlson et al. (1987) suggests that this negative radiative
forcing will have cascading effects on marine primary productivity, leading to a DMS-
mediated climate feedback loop. Since its publication in 1987, the CLAW hypothesis has
provided motivation for the widespread measurement of DMS in the global ocean.
Beyond their potential role in regional climate forcing, DMS and DMSP also play
critical ecological roles in marine microbial metabolism and food-web dynamics (for a
complete overview; see Stefels et al., 2007). DMSP is believed to serve numerous
physiological functions in phytoplankton, with suggested roles as an osmolyte, an anti-
oxidant, and a cryoprotectant under different environmental conditions. Sunda et al.
(2002) suggested that oxidative stressors, such as high solar radiation or iron limitation,
may stimulate DMSP production in certain phytoplankton species. The production of this
molecule is largely species-dependent, and can vary by three orders of magnitude among
phytoplankton groups, with the highest intracellular concentrations typically reported in
dinoflagellates and haptophytes, and lower concentrations in diatoms (Keller, 1989).
After synthesis, DMSP can be cleaved to DMS and acrylate within algal cells, or

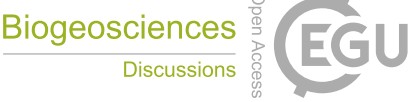

by heterotrophic bacteria acting on the dissolved DMSP ($DMSP_d$) pool in the water
column (Zubkov et al., 2001).  The release of DMSP into the water column is believed to
be enhanced in physiologically stressed or senescent phytoplankton (Malin et al., 1998).
and can be stimulated by zooplankton grazing and viral lysis (Zubkov et al., 2001).
Bacteria can also utilize $DMSP_d$ as a sulfur source for protein synthesis (Kiene et al.,
2000), but this pathway does not lead to DMS release.  The DMS yield of bacterial
DMSP metabolism (i.e. the fraction of consumed DMSP that is converted to DMS) varies
significantly, and may be influenced by the relative supply and demand of reduced sulfur
and carbon for bacterial growth (Kiene and Linn, 2000).

Modeling studies have suggested that DMS emissions could exert an especially

significant influence on regional climate in polar regions, due to the low background
concentrations of atmospheric aerosols at high latitudes (Woodhouse et al., 2010). In
support of this, direct observations have demonstrated a link between particle formation
events in the Arctic atmosphere and sea surface DMS emissions (Chang et al., 2011),
(Mungall et al., 2016), motivating further quantification of marine DMS emissions in
Arctic regions. Yet, logistical constraints have limited the measurements of surface water
properties in many high latitude regions, and these areas remain relatively sparsely
sampled for DMS/P concentrations.  Indeed, of the approximately 50,000 data points in
the global Pacific Marine Environmental Laboratory (PMEL) database of oceanic DMS
measurements (http://saga.pmel.noaa.gov/dms/), only 5 % have been made in either
Arctic or Antarctic waters (~ 1600 and 1000 data points, respectively).

Despite the relatively limited sulfur observations in high latitude waters, an





examination of the available data reveals large differences in the water column DMS
distributions of the Arctic and Antarctic regions. While the summertime mean DMS
concentration in the Arctic Ocean is 3.0 nM (close to the global mean value of 4.2 nM,
derived from the PMEL data), the mean summertime DMS concentration in the Southern
Ocean is ~ 3 times higher at 9.3 nM. Moreover, several areas of extraordinarily high
DMS concentrations (>100 nM) have been observed in various regions of the Southern
Ocean (DiTullio et al. 2000; Tortell et al. 2011), whereas no study to date has observed
DMS concentrations above 25 nM in Arctic waters.  The available data thus suggest
contrasting dynamics of DMS/P production in the two polar regions (i.e. Arctic vs.
Antarctic).

Although Arctic and Antarctic regions share several key physical characteristics,

most notably strong seasonal cycles in sea ice cover and solar irradiance, there are some
critical differences.  Much of the pelagic Southern Ocean is an iron-limited, High
Nutrient Low Chlorophyll (HNLC) regime, with strong seasonal changes in mixed layer
depths (Boyd et al., 2001).  Low iron conditions, and seasonally-variable mixed layer
light levels may induce oxidative stress (particularly in ice-influenced stratified waters)
and thus promote high DMS production (Sunda et al., 2002).  In addition, parts of the
Southern Ocean are characterized by extremely high biomass of *Phaeocystis antarctica*
(Smith et al., 2000), a colonial haptophyte that is a prodigious producer of DMSP and
DMS (Stefels et al., 2007).  By comparison, the salinity-stratified surface waters of the
Arctic Ocean are believed to be primarily limited by macronutrient  (i.e. nitrate)
availability (Tremblay et al., 2006), with a maximum phytoplankton biomass that is at





least an order of magnitude lower than that observed in the Southern Ocean (Carr et al.,
2006).  Despite the relatively low phytoplankton biomass over much of the Arctic Ocean,
reasonably high summertime DMS levels (max ~ 25 nM) have been observed in some
regions. It is also important to note that significant Arctic phytoplankton biomass and
primary productivity may occur in sub-surface layers (Martin et al. 2010), and in under-
ice blooms (Arrigo et al., 2012).  The quantitative significance of these blooms for DMS
production is unknown at present (Galindo et al., 2014).

Quantifying the spatial and temporal distribution of DMS and DMSP in the Arctic

Ocean is particularly important in light of the rapidly changing hydrographic conditions
across this region. Rapid Arctic warming over the past several decades has been
associated with a significant reduction in the extent of summer sea ice, resulting in higher
mixed layer irradiance levels and a longer phytoplankton growing season (Arrigo et al.,
2008). Arrigo et al (2008) suggested that continued warming and sea-ice loss could lead
to a three-fold increase in primary productivity over the coming decades.  The effects of
these potential changes on DMS/P concentrations and cycling remain unknown, but it has
been suggested that future changes in Arctic Ocean DMS emissions could modulate
regional climatic patterns (Levasseur, 2013).  Indeed, modeling work has suggested that
cooling associated with increased DMS production and emissions in a less ice-covered
Arctic may help offset warming associated with loss of sea-ice albedo (Gabric et al.,
2004). The important climatic and biological roles of reduced sulfur compounds,
combined with altered marine conditions under a warming environment, provide the
motivation for a deeper understanding of the distribution and cycling of DMS and related



123 compounds in Arctic waters.

124   In this article, we present a new data set of DMS and DMSP concentrations in

125 Arctic and Subarctic waters adjacent to the Canadian continental shelf. We used a

126 number of recent and emerging methodological approaches to measure these compounds

127 in a continuous ship-board fashion. In particular, we used membrane inlet mass

128 spectrometry (MIMS) to measure DMS with extremely high spatial resolution (i.e. sub-

129 km scale), and the recently developed organic sulfur sequential chemical analysis robot

130 (OSSCAR), for automated analysis of DMS and DMSP. Our goal was to utilize the

131 sampling capacities of the MIMS and OSSCAR systems to make simultaneous

132 measurements of DMS/P in Subarctic Atlantic and Arctic waters, in order to expand the

133 spatial coverage of the existing DMS/P dataset, and identify environmental conditions

134 leading to spatial variability in the concentrations of these compounds.

135 **2. Methods**

136 **2.1 Study Area**

137 Our field study was carried out on board the *CCGS Amundsen* during Leg 2 of the 2015

138 GEOTRACES expedition to the Canadian Arctic, (July10 – August 20, 2015). We

139 sampled along a ~ 10,000 km transect from Quebec City, Quebec, to Kugluktuk,

140 Nunavut. Data collection commenced off the coast of Newfoundland, and included

141 waters of the Labrador Sea, Baffin Bay and the Canadian Arctic Archipelago (Fig. 1).

142   The cruise transect covered two main distinct geographic domains – the Baffin

143 Bay/Labrador Sea region, and the Canadian Arctic Archipelago (CAA). The majority of

144 the surface water in the Canadian Arctic Archipelago is from Pacific-sourced water





masses, as a shallow sill near Resolute limits the westward flow of Atlantic-sourced water
(Michel et al., 2006). Flow paths through the CAA are complex. The region is
characterized by a network of shallow, narrow straits that are subject to significant
regional variability in local mixing and tidal processes, and strongly influenced by
riverine input, which drives stratification (Carmack et al, 2011).  By contrast, both
Atlantic- and Pacific-sourced waters mix in the Baffin Bay and Labrador Sea regions, and
this confluence drives a strong thermohaline front. These regions are less strongly
stratified than the CAA (Carmack et al, 2011).
**2.2 Underway sampling systems**
We utilized two complementary underway sampling systems to measure reduced
sulfur compounds; membrane inlet mass spectrometry (MIMS; Tortell, 2005)), and the
organic sulfur sequential chemical analysis robot (OSSCAR; Asher et al., 2015)).
Detailed methodological descriptions of these systems have been published elsewhere
((Tortell, 2005, 2011), (Asher et al., 2015)), and only a brief overview is given here.
**2.2.1 OSSCAR**
The OSSCAR instrument consists of an automated liquid handling / wet
chemistry module that is interfaced to a custom-built purge-and-trap gas chromatograph
(GC) equipped with a pulsed flame photometric detector (PFPD) for sulfur analysis.  A
custom LabVIEW program is used to automate all aspects of the sample handling and
data acquisition. During analysis, unfiltered seawater (3 - 5 ml) from an underway supply
(nominal sampling depth $\sim$ 5 m) is drawn via automated syringe pump into a sparging
chamber.  DMS is then stripped out of solution (4 minutes of 50 ml min$^{-1}$ N$_2$ flow) onto a



1/8" stainless steel trap packed with carbopack at room temperature. Rapid electrical
heating of the trap (to ~260°C), causes DMS desorption onto a capillary column (Restek
SS MXT, 15m, 80 °C, 2 ml min$^{-1}$ N$_2$ flow) prior to detection by the PFPD (OI Analytical,
Model 5380).  Light emitted during combustion in the PFPD is converted to a voltage and
recorded by a custom built Labview data acquisition interface. Following the completion
of DMS analysis, 5 N sodium hydroxide is added to the sparging chamber for 14 minutes
to cleave DMSP in solution to DMS, following the method of Dacey and Blough (Dacey
and Blough, 1987).  The resulting DMS is sparged out of solution and measured as
described above.  The sparging chamber is then thoroughly rinsed with Milli-Q water,
and the process can be repeated.  As we used unfiltered seawater for our analysis, it is
important to note that we measured total DMSP (DMSP$_t$) concentrations, which represent
the sum of dissolved and particulate pools.

We measured an in-line standard every 4-5 samples (at most every 3 hours) to

ensure that the system was functioning correctly, and to correct for potential detector
drift.  The mean standard error of daily point standards was 0.55 nM, and we consider
this to represent the precision of our emerging method (significant efforts are underway
to increase this precision). To correct the underway data for instrument drift, point
standard measurements were smoothed with a 3-pt running mean filter, interpolated to the
time-points of sample measurements, and compared to the known standard concentration
to provide a drift correction factor for every seawater data point.  Six-point calibration
curves were performed every two days, using DMS standards (ranging from 0 to 18nM),
produced from automated dilutions of a primary DMS stock and Milli-Q water (see Asher





et al., 2015). The limit of detection of the system was calculated from the calibration
curve using the formula $C_{LOD} = 3s_{y/x} \div b$, where $C_{LOD}$ is the concentration limit of
detection, $s_{y/x}$ is the standard error of the regression , and b is the slope of the regression
line.  With this approach, we derived a mean limit of detection of 1.4 nM.  The mean
linear calibration curve $R^2$ value, taken over all calibration curves, was 0.9887.

The OSSCAR system is designed to automate the collection of seawater for

sequential analysis of DMS, DMSO, and DMSP in a single sample.  During our cruise,
however, we experienced problems with the DMSO reductase enzyme used to convert to
DMS for analysis, and we therefore configured the instrument to run only DMS and
DMSP at sea, with one cycle requiring roughly 30 minutes.
**2.2.2 MIMS**
We used Membrane Inlet Mass Spectrometry (MIMS) to obtain very high frequency
measurements (~ several data points per minute) of DMS concentrations and other gases
in surface seawater.  Using this system, seawater from the ship's underway loop was
pumped through a flow-through sampling cuvette, attached, via a silicone membrane, to a
quadrupole mass spectrometer (Hiden Analytical HPR-40).  DMS was measured by
detecting ions with a mass to charge ratio of 62 (m/z 62) every ~30 seconds.  To achieve
constant sample temperature prior to contact with the membrane, seawater was passed
through a 20 foot coil of stainless steel tubing immersed in water bath held at 4 °C
(Tortell et al. 2011). The system pressure (as measured by the Penning Gauge) remained
stable during operation ($\sim 1.3 - 1.5 \times 10^{-6}$ Torr). The DMS signal was calibrated using
liquid standards that were produced by equilibrating 0.2 $\mu$m filtered seawater with a



constant supply of DMS (m/z 62) from a calibrated permeation device (VICI  Metronics).
The primary effluent from the permeation tube (held at 30 °C) was split among several
capillary outflows and mixed into a $N_2$ stream (~ 50 ml min$^{-1}$) to achieve a range of
DMS / $N_2$ mixing ratios for bubbling into standard bottles held in an incubator tank
supplied with continuously flowing seawater.  Concentrations of DMS in the standard
bottles were cross-validated by measuring discrete samples using the OSSCAR system.
**2.3 Post-processing of DMS data**
Raw data outputs (voltages) for both OSSCAR and MIMS measurements were processed
into final concentrations using MATLAB scripts.  For OSSCAR data, raw voltages were
captured with a sampling frequency of 5 Hz.  Sulfur peaks eluting off the GC column
were integrated using a custom MATLAB script, with correction for baseline signal
intensities.   DMS concentrations were derived from peak areas using the calibration
curves as described above.
**2.4 Ancillary seawater data**

Shipboard salinity, temperature, wind speed, and chlorophyll *a* (chl *a*)

fluorescence measurements were collected using several underway instruments. We used
a Seabird Electronics thermosalinograph (SBE 45) for continuous surface temperature
and salinity measurements, and a Wetlabs Fluorometer (WetStar) to measure chl *a*
fluorescence, as a proxy for phytoplankton biomass.  We note that the chl *a* fluorescence
data are subject to significant diel cycles associated with light-dependent fluorescence
quenching. All sensors were calibrated prior to and following the summer expedition.
Conductivity Temperature Depth (CTD) profiles were used to measure vertical profiles of



salinity and potential temperature at 17 stations, from which we computed density using
the Seawater Toolbox in MATLAB.  The mixed layer depth was defined as the depth
where density exceeded surface values by 0.125 kg m$^{-3}$.  Sea ice concentrations were
obtained from the AMSR-E satellite product (Cavelieri et al. 2006) with a spatial
resolution of 12.5 km.  The percent ice cover along the cruise track was derived from a
two dimensional interpolation of the ship's position in time and space against the daily
sea ice data.
All correlation analyses (Pearson's *r*) were computed in MATLAB, using the
corrcoef function. Sample sizes were as follows: 33,250 data points in the MIMS DMS
dataset, 344 in the OSSCAR DMS dataset, and 318 in the OSSCAR DMSP dataset.
**2.5 Phytoplankton biomass and taxonomic composition**
In addition to underway data, samples for the quantification of photosynthetic and
accessory pigments (Table 1) were collected at a number of discrete oceanographic
stations (see Table 2). For each station, duplicate samples (250-500 mL) for chl *a* analysis
were filtered onto pre-combusted 25 mm glass fiber filters (Whatman GF/F) using low
vacuum pressure (<100 mm Hg). Filters were stored at -20 ºC and chl *a* was determined
within a few days of sample collection using fluorimetric analysis following the method
of Welschmeyer (Welschmeyer 1994). Duplicate 1-2 L samples were filtered onto pre-
combusted 25 mm GF/F for pigment analysis by reverse-phase High-Performance Liquid
Chromatography (HPLC). Filters were dried with absorbent paper, flash frozen in liquid
nitrogen and stored at -80 ºC until analysis following the method of Pinckney et al
(1994). We used several diagnostic pigments as markers for individual phytoplankton



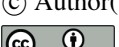

groups, as described by Coupel et al (2015) (see Table 1). Following HPLC pigment
processing, data were interpreted with the chemotaxonomy program CHEMTAX V1.95,
using the pigment ratio matrix described by Taylor et al (2013).
**2.6 DMS Sea-Air Flux**
We derived sea-air fluxes of DMS from MIMS measurements of DMS
concentrations, as these data had higher resolution and spatial coverage than OSSCAR
observations. We computed sea-air flux as:
$$F_{DMS} = k_{DMS} \, (DMS_{SW}) \, (1 - A)^{0.4} \tag{1}$$
Where $DMS_{sw}$ is the concentration of DMS in the surface ocean and $k_{DMS}$ is the gas
transfer velocity derived from the equations of Nightingale et al. (2000), normalized to
the temperature and salinity-dependent DMS Schmidt number of Saltzman et al. (1993).
The term A represents percent sea ice cover, and the scaling exponent of 0.4 accounts for
the effects of sea ice on gas exchange and is derived from the work of Loose et al. (2009).
Sea surface salinity and temperature measurements described in section 2.5 were used in
the calculations. Wind speed data were obtained from the ship's anemometer (AAVOS
data, Environment Canada).
**3. Results**
**3.1 Oceanographic setting**
Figures 1 and 2 show the distribution of hydrographic properties across our cruise
survey region. Over our sampling area, surface water temperatures varied between -1.2
and 10.2 °C, while surface salinity ranged from 10.7 to 34.7 psu (Fig. 1). The warmest
and most saline waters were found in the Labrador Sea, with cold fresher waters in



Hudson Strait and the Canadian Arctic Archipelago. Underway chl *a* fluorescence varied
between 0.04 and 2.96 μg L$^{-1}$, averaging 0.20 μg L$^{-1}$.  The highest chl *a* fluorescence was
observed in a localized region within Baffin Bay, in the vicinity of a sharp temperature
and salinity frontal zone (Fig. 1). Mixed layer depths ranged from ~ 5 - 50 m, and were
deepest in the Labrador Sea and shallowest in the stations of the Canadian Arctic
Archipelago.  Sea ice cover was variable across the survey transect, with ice-free waters
in the Labrador Sea, and significant ice cover in the northern Hudson Bay and parts of the
Canadian Arctic Archipelago (Fig. 2).
**3.2 Phytoplankton biomass and taxonomic distributions**
Using measurements of accessory photosynthetic pigments, we examined spatial patterns
in the taxonomic composition of phytoplankton assemblages (see Table 1 for a
description of HPLC marker pigments and their associated phytoplankton taxa).The
distribution of pigments across our sampling stations is presented in Table 2, along with
measurements of mixed layer depth and ice cover, while CHEMTAX-derived assemblage
estimates are shown in Table 4. In order to remove large potential differences in total
phytoplankton biomass, we normalized pigment concentrations to total chl a
concentrations measured using HPLC (see Methods, section 2.5).
CHEMTAX pigment analysis shows that all stations in the study area were diatom-
dominated, although haptophyte, dinoflagellate, and prasinophyte markers were detected
in varying quantities at all stations (see Table 4).  Total HPLC-measured chl *a* was
relatively low throughout the study area, ranging from 0.11 to 0.56 μgL$^{-1}$.
**3.3 Observed DMS/P concentration ranges**





The DMS data shown in Fig. 1 are derived from MIMS measurements, since

these have wider geographic coverage and greater spatial resolution than OSSCAR data.
DMS concentrations measured with MIMS ranged from 0.2 nM to 12 nM, averaging 2.7
(± 1.5) nM.  The highest values were observed in the northern Labrador Sea, Baffin Bay
and Hudson Strait, with lower values through much of the Arctic Archipelago.

Figure 3 shows the distribution of DMS, measured by both MIMS and OSSCAR,

along the cruise track.  DMS concentrations measured with OSSCAR ranged from 0.1 to
18nM, averaging 3.2± 2.4nM.  In general, we observed reasonably good coherence
between DMS measurements made by our two analytical systems, with similar absolute
values of data and spatial patterns.  There were, however, notable offsets in the early
August measurements (~ km 7000 cruise track, Fig. 3a), when OSSCAR DMS data were
consistently higher than MIMS data.  Notwithstanding this offset (for which potential
reasons are addressed in the discussion), the good coherent spatial patterns in data
derived from these independent methods is encouraging, particularly given the rather low
precision of our current OSSCAR system.

The spatial distribution of DMSP concentrations (measured with OSSCAR) along

the cruise track is also shown in Fig. 3.  Concentrations ranged from <1 nM to 160 nM,
and averaged 30 ± 29nM.  DMSP:chl $a$ ratios measured from HPLC chl a data ranged
from 52.31nmol μg$^{-1}$ to 181.4nmol μg$^{-1}$.  Examination of the data in Figure 3
reveals that high DMS concentrations were sometimes, but not always, accompanied by
high DMSP concentrations.  For example, a sharp increase in measured DMSP
concentrations (around  7000-7400 km) on the cruise track was accompanied by a sharp





increase in DMS measured by both instruments, while low-DMS waters observed around
km 9400 along the transect also showed very little DMSP.  Over the portion of the
transect where measurements of both DMS and DMSP were available, the OSSCAR-
measured concentrations of these compounds exhibited a statistically significant positive
correlation (r = 0.52, p< 0.001). There were, however, a number of regions where
increased DMS concentrations were not accompanied by increases in DMSP (e.g. ~ km

10,000).

**3.4 Sea-Air Flux**
Figure 5 shows DMS sea-air fluxes as computed from MIMS-measured DMS seawater
concentrations, wind speed and sea ice cover. DMS sea-air fluxes ranged from < 1 to 80
$\mu$mol S m$^{-2}$ day$^{-1}$, with peak sea-air flux observed around km 5500 on the cruise track.
Sea-air flux is highly dependent on wind speed and sea ice cover, with the result that even
high concentrations of seawater DMS yielded low sea-air flux when low wind and/or
high sea ice was present (e.g. km 2100, 7200, 8300). Conversely, very high sea-air fluxes
were observed when moderately high DMS concentrations coincided with high wind
speeds and ice-free waters (e.g. km 5400).
**3.5 Comparison of gradients in DMS data with hydrographic features**
The high sampling frequency of MIMS measurements allows the comparison of
DMS observations with other underway environmental variables, and enables the
quantification of small-scale DMS concentration gradients in near real-time. Figure 2
shows a cruise track record of MIMS-measured DMS concentrations in relation to
salinity, temperature, chl *a* fluorescence, and ice cover. Several sharp increases in DMS at



around kms 2100, 3300, and 3800 along the cruise track were accompanied by strong
gradients in temperature and, to a lesser extent, salinity (Fig. 2). These regions
correspond to areas in the Labrador Sea and Baffin Bay. An increase in DMS
concentrations in Baffin Bay around km 7200 in the cruise track (Fig 2a) was associated
with a simultaneous drop in sea-surface temperature and salinity, in close proximity to a
sharp increase in chl *a* fluorescence along the cruise track (Fig 2c) (see Fig. 1). As shown
in Fig 3b, this localized region exhibited the highest concentrations of DMSP along the
transect. Interestingly, this area was also characterized by strong gradients in sea ice
concentrations, and the low salinity waters are indicative of localized ice melt. Figures 1d
and 2d also show the large-scale salinity gradients in the Hudson Bay and the Canadian
Arctic Archipelago, highlighting the freshwater influx in these near-shore areas.  In
contrast to our observations in Baffin Bay, DMS concentrations showed relatively little
variability across these salinity gradients.

In order to more closely examine small-scale variability in DMS and other surface

water variables, we calculated spatial gradients in the data to examine the coherence of
frontal features in DMS, salinity, temperature and chl *a* fluorescence. For this analysis,
we computed gradients in each oceanographic variable within a neighborhood of 100
points surrounding each point. Gradients (G) for each variable (DMS, SST, chl *a*, and
salinity) were calculated at each point x as follows:
$$G_x = \frac{V_{x+50} - V_{x-50}}{D_{x+50} - D_{x-50}} \quad (2)$$

Here, G is gradient (in units of change per km), V is the value of the variable at a

point, x, and D is the cruise track distance at x. A neighborhood of 100 points was



subjectively chosen because it best captured the observed variability in the data,
representing an intermediate value between a localized neighborhood (e.g. 10 points),
which would only consider changes close to the point, and a large neighborhood (e.g.
1000 points), which would smooth the features. The results of this analysis (Fig. 4)
qualitatively demonstrate a coherence of DMS gradients with salinity, chlorophyll, and
sea surface temperature.
**3.6 Correlation with ancillary oceanographic variables**

We computed Pearson correlation coefficients of DMS and DMSP with underway

measurements of salinity, sea surface temperature, chl $a$ fluorescence, and sea ice cover.
We also examined the potential relationships between DMS concentrations and MIMS-
derived $pCO_2$, and $\Delta O_2/Ar$ (Tortell et al., in preparation). The results can be seen in Table
3. Only correlations significant at the 0.05 level are included. Only weak correlations are
seen between MIMS-measured DMS data and ancillary variables, and OSSCAR DMS
data did not exhibit any significant correlations with any ancillary variables, including
measured of phytoplankton taxonomic distributions. A strong positive correlation ($r =$
0.66, $p<0.001$) was found between DMSP and underway chl $a$ fluorescence. Over the
whole transect, we observed a weak negative correlation between DMS/P and sea-ice
cover ($r = -0.26$ for DMS, and $r=-0.34$ for DMSP, $p< 0.001$ in both cases). A weak
positive correlation was found between DMSP/chl $a$ and ice cover ($r = 0.52$, $p< 0.04$),
suggesting potential roles for sea-ice microalgae in DMSP production at the sampled
stations. It is interesting to note that elevated chl $a$ fluorescence and DMSP
concentrations often occurred in areas of intermediate ice cover (km 3300, 7300 and 9200



along the cruise track), potentially reflecting the influence of ice-edge blooms or under-
ice phytoplankton assemblages. Potential mechanisms for these features are addressed in
the discussion.

**4. Discussion**

Our results provide a new dataset of reduced sulfur compounds in an under-

sampled region of the Arctic Ocean, and enable an examination of DMS/P variability in
relation to a number of oceanographic properties on a range of spatial scales.  Below, we
focus our discussion on the observed relationship between gradients in DMS and other
oceanographic variables, and discuss the comparability of the two DMS measurement
methods utilized.   We compare our results to previously published measurements in the
Arctic, situating our results in the context of the changing hydrography and
phytoplankton ecology of the Arctic Ocean.
**4.1 Comparability of MIMS and OSSCAR measurements**

The OSSCAR and MIMS instruments have previously shown good agreement in

measured DMS concentrations in the Subarctic Pacific Ocean (Asher et al. 2015).
Similarly, we observed relatively good coherence between the two methods (Fig. 3) over
much of our cruise track.  The largest exception to this occurred around km 7000, when
DMS measurements measured by OSSCAR were significantly higher than those
measured by MIMS. This region was characterized by very high DMSP measurements
(often one order of magnitude higher than the DMS measurements).  If small amounts of
DMS remained in the OSSCAR system after DMSP analysis, sample carry-over could
contribute to higher measured concentrations in the subsequent DMS analysis.  In order



to minimize this potential artifact, the system was thoroughly rinsed with MilliQ water
after every run.  It is possible, however, that this approach was not entirely efficient.
Another potential cause of the higher OSSCAR DMS measurements may be due to cell
breakage during the sparging process in OSSCAR.  In this scenario, there is the potential
for release of intracellular DMSP and DMSP lyase into solution, which would lead to
artificially high measured DMS concentrations.  It is not possible for us to quantify the
magnitude of such a potential artefact, but we note that its magnitude would likely
depend on the taxonomic composition of phytoplankton assemblages.  Wolfe et al (2002)
showed that sample sparging led to an increase in DMS production by both the
haptophyte *Emiliana huxleii*  and the dinoflagellate *Alexandrium.* (Wolfe et al, 2002).
Unfortunately, due to limited coverage of discrete sampling, we do not have any
estimates of phytoplankton community composition in the region where MIMS and
OSSCAR showed the greatest discrepancies. Notwithstanding these potential caveats, we
suggest that the two methods show strong promise to provide complementary information
on DMS/P (and DMSO) concentrations in surface ocean waters.

One challenge going forward is to increase the reproducibility of OSSCAR

measurements, and this is an area of active work in our group.  Moreover, we have
recently worked to significantly improve the limit of detection.  The version of our
system used in 2015 had a detection limit of roughly 1.4 nM, and was thus far less
sensitive than many conventional GC methods, which can achieve sub-nM detection
limits.  Our detection limit was of only minor consequence for DMSP measurements,
given that 72% of measured DMSP concentrations were higher than 10 nM, and less than



3% fell below 1.4 nM. The relatively low sensitivity was somewhat more problematic
for DMS, with approximately 22% of our OSSCAR-measured DMS values below 1.4
nM. Nonetheless, as discussed below, we believe that the OSSCAR data, in combination
with our MIMS data, provide useful information on the spatial distribution of both DMSP
and DMS in Arctic waters.
**4.2 Towards a regional Arctic data base of DMS/P concentrations**

Figure 6 shows a comparison between our Arctic DMS measurements (made by

OSSCAR) and other summertime Arctic DMS data in the PMEL database. For this
comparison, only PMEL measurements made above the Arctic circle (66.56° N) in June-
August were included, resulting in a total of 415 data points. As shown in Fig. 6, the
majority of available summertime PMEL DMS/P measurements are found in the Atlantic
region of the Arctic, and in the Bering Sea, with limited data in the Canadian Archipelago
(for an overview of Arctic DMS/P studies performed to date, see Levasseur, 2013 ). For
the sake of visual clarity, the presentation of data in Fig. 6a, is based on DMS
measurements made by OSSCAR, whereas both sets of data were included in the
frequency distribution analysis (Fig. 6b).The results presented in Fig. 6 suggest that our
measurements are representative of the broader Arctic context, with generally similar data
frequency distributions (Fig. 6b) for all three DMS datasets (MIMS, OSSCAR, and
PMEL). From the map, we see that the spatial footprint of our measurements complement
the existing summer data, helping to expand the spatial coverage of DMS observations in
the Arctic Ocean. While the PMEL data base does not include information needed to
directly calculate sea-air fluxes, the range of sea-air fluxes we calculated ($\sim 1 - 80$ μmol



m$^{-2}$ d$^{-1}$) was consistent with recent summertime sea-air DMS fluxes modeled in Resolute
Bay (Hakase Hayashida, pers. comm.).

In addition to complementing the existing PMEL DMS database, our new

observations also build on a number of other reduced sulfur measurements in the
Canadian Sector of the Arctic Ocean. Observations of DMS and DMSP derived from
several past Arctic and subarctic Atlantic surveys are summarized in Table 5. This table
focuses heavily on DMS and DMSP measurements made in the Canadian sector and
Greenland waters, serving to provide context for our measurements performed in similar
environments.  The data presented in Table 5 are drawn from different times of year, and
from phytoplankton assemblages of varying taxonomic composition, allowing us to
examine sulfur accumulation in surface waters under a range of environmental and
ecological conditions.   For example, Bouillon et al. (2002) observed low DMS
concentrations (<1nM) during a large spring diatom bloom (~ 15 μg L$^{-1}$chl *a*) in the North
Water region.  In contrast, higher DMS concentrations have been reported later in the
season when total phytoplankton biomass is lower, and taxonomic composition has
shifted away from diatom-dominance. Working in the same geographic region as
Bouillon, Motard-Côté et al. (2012) reported higher late summer (September) DMS
levels (maximum = 4.8nM), which were accompanied by moderate chl *a* concentrations
(0.2-1 μgL$^{-1}$ ), while Luce et al. (2011) reported very low DMS (<1nM) associated with
moderate chl *a* concentrations (0.2-2 μgL$^{-1}$) in a flagellate dominated community in late
fall (October-November), with DMS decreasing towards the later months.  A similar
pattern was observed in the Northwest Subarctic Atlantic by Lizotte et al (2012), who





associated elevated reduced sulphur (DMSP) production with flagellate and
prymnesiophyte communities in midsummer and fall, in contrast to early-season diatom
blooms with little associated DMSP and DMS. This seasonal decrease in DMS levels
may be potentially attributable to light limited primary productivity, and diminishing
capacity for light-induced oxidative stress, which has been shown to increase DMS/P
production (Sunda et al., 2002).

To date, the highest recorded Arctic water column measurements of DMS (25nM)

and DMSP (160 nM) have been observed during mid-summer blooms of the haptophyte
*Phaeocystis* at the ice edge (see Matrai and Vernet, 1997; Gali and Simo, 2010). Our mid-
season (July-August) study of similar areas shows moderately high DMS (up to 18 nM)
accompanied by relatively low chl *a* (0.11- 1.06 μgL$^{-1}$) in a mixed community where
flagellates and prasinophytes  are present (see discussion of HPLC pigments).

Together, the available data (Table 5 and our measurements) are consistent with a

seasonal cycle in Arctic and subarctic reduced sulfur distributions. Early season diatom-
dominated blooms exhibit high biomass and primary productivity but low DMS/P
accumulation, while mid-summer phytoplankton assemblages dominated by haptophytes
and dinoflagellates display lower phytoplankton biomass but higher reduced sulfur
accumulation.  This pattern is similar to the summertime 'DMS paradox' in lower latitude
temperate and sub-tropical marine waters (Simo and Pedrós-Alió, 1999).  In the fall, both
Arctic primary productivity and DMS/P production decrease with the onset of lower
temperatures and increased ice cover.  Our data are consistent with this general scenario,



representing a mixed-species assemblage with moderate biomass and DMS/P
accumulation.
**4.3 Gradients in DMS and hydrographic frontal structures**
The high resolution afforded by the MIMS dataset allows for the observation of
fine-scale variability in DMS concentrations at the sub-kilometer scale. Previous studies
(Tortell, 2005; Tortell et al., 2011) have previously quantified fine-scale variability in
DMS concentrations, demonstrating de-correlation length scales on the order of 10s of
Km, and often shorter than that of other oceanographic variables such as temperature and
salinity. Figures 2 and 4 clearly demonstrate that gradients in DMS and chl *a*
fluorescence often co-occur with strong gradients in temperature and salinity. This
suggests a potential role for hydrographic fronts in driving changes in DMS
concentrations. Several potential mechanisms may explain this phenomenon. For
example, the frontal mixing of distinct water masses, driven by currents, wind, or melting
ice, may introduce nutrients into a low-nutrient water column, stimulating primary
productivity and potentially increasing DMS/P production. This stimulation of primary
productivity has been observed previously by other groups. For example, Tremblay et al.
(2011) showed that introduction of nutrient-rich water masses through ice ablation and
upwelling led to large (2-6 fold) increases in phytoplankton primary productivity
(Tremblay et al., 2011). Mixing of water masses may also potentially expose water
column phytoplankton to light shock or osmotic stress by mixing them upwards in the
water column or introducing an abrupt salinity gradient. Both of these factors could
contribute to elevated DMSP production, given its hypothesized role as an intracellular



osmolyte and anti-oxidant (Stefels et al., 2007). Though our data do not allow
mechanistic interpretation for the underlying causes of DMS variability in surface waters,
the high resolution afforded by MIMS measurements enables real-time observations of
DMS gradients, which may be useful in the design of future process studies examining
the driving forces for elevated DMS accumulation.

Fully resolving the production and consumption dynamics of DMS/P in seawater

requires a series of time-consuming and laborious methods, including various isotope
tracer studies and quantification of multiple physical process rates (e.g. photo-oxidation).
Clearly, it is not possible to conduct such measurements with the high frequency of our
MIMS-based DMS measurements.  However, use of real-time MIMS monitoring will
enable the selection of targeted sampling locations to best leverage sampling efforts.  In
this respect, the recent work of Asher et al. (2016) provides some example of how high
resolution DMS/P measurements can be coupled with isotope tracer studies to derive
insight into DMS/P dynamics in high latitude marine waters.
**4.4 Phytoplankton assemblage composition and mixed layer depth**

The majority of the sampled stations were characterized by very shallow mixed

layer depths (MLD; Table 2) resulting from strong salinity-based stratification of surface
waters. Light stress associated with shallow MLD may contribute to elevated DMSP : chl
*a* ratios.  In our dataset, the shallowest MLDs were observed at stations BB3 and CAA6
(8.2 m and 6.1 m, respectively), and these stations were also characterized by elevated
DMSP concentrations. The elevated DMSP : chl *a* ratios measured in our study also
reflect the presence of high-DMSP producing taxa, a phenomenon also reported by other





groups (Matrai et al. 1997; Gali et al., 2010; Lizotte et al., 2012). Limited HPLC station
data suggest that a mixed phytoplankton assemblage was present in the study area at the
time of sampling.  When comparing our DMSP: chl *a* ratios to other measurements, it is
important to note that we measured $DMSP_t$, while many other groups present results in
terms of $DMSP_p$, without taking into account the dissolved fraction ($DMSP_d$).  As the
dissolved DMSP pool typically makes up a small (though highly variable) portion of the
total water column DMSP pool, the use of DMSPt does not likely have a large effect on
derived DMSP:chl a ratios (Kiene et al., 2000; 2006).  Moreover, we used HPLC-derived
chl a for these calculations, as opposed to the more standard fluorometric chl a
measurements.  HPLC chl a measurements are likely to be more accurate than
fluorometric measurements, and tend to yield lower concentrations, acting to increase
DMSPt:chl a (Welschmeyer 1994) .

Despite the potential caveats raised above, the $DMSP_t$:chl *a* ratios we measured

across our sampling stations (52-182 $nM\mu g^{-1}$ ) were broadly similar to $DMSP_p$:chl *a*
values found by Motard-Côté et al. (15-229 $nM\mu g^{-1}$) in the same region in September
(Motard-Côté et al., 2011). In contrast, our measured $DMSP_t$:chl *a* ratios are significantly
higher than those measured by Luce et al. (maximum of 39 $nM\mu g^{-1}$) (Luce et al., 2007)
and Matrai and Vernet (maximum 17 nmol $\mu g^{(-1)}$) at diatom-dominated stations in the
Barents Sea (Matrai and Vernet., 1997).  This difference likely reflects a difference in
phytoplankton assemblage composition, even though we were unable to find any
significant correlations between DMSP:Chl and HPLC pigment markers for different
phytoplankton groups.  It may be that the taxonomic composition of our sampled





assemblages was not sufficiently variable to enable large differences in DMSP:chla
values.

### 4.5 The interaction of DMS/P and sea ice

The presence of sea ice exerts a strong control on polar phytoplankton by limiting
irradiance for primary productivity in the water column. This allows high concentrations
of nutrients to accumulate, creating favorable conditions for phytoplankton blooms upon
sea-ice melt.  Ice edge blooms are well documented, and can serve as a source for
reduced sulfur compounds. In a 2010 study, Gali et al found that sea ice melt drove
stratification of nutrient rich surface water, triggering a sharp increase in primary
productivity, with associated elevated DMS and DMSP levels (Gali et al., 2010).  A
number of recent studies have also examined the potential of sea ice to act as a reservoir
of reduced sulfur.  For example, Levasseur et al (1994) reported very high concentrations
of DMS and DMSP in Arctic bottom-ice diatoms, and suggested that the breakup of sea
ice may stimulate reduced sulfur production by triggering phytoplankton blooms and
releasing accumulated sulfur into the water column. In a more recent study , Galindo et al
(2016) demonstrated experimentally that the exposure of phytoplankton to high light
conditions (mimicking those that would follow the breakup of sea ice)  led to near-total
release of intracellular DMSP, providing one possible explanation for elevated DMSP
levels in the water column.
The weak negative correlations between sea ice cover and DMS/P concentration
we observed is consistent with the idea that sea ice cover limits insolation, thereby
reducing primary productivity and DMS/P production. In general, the drivers of DMSP





and DMS production differ – though DMSP production has been shown to be directly
influenced by sea ice melt in under-ice blooms [30a], the production of DMS from DMSP
is largely dependent on the metabolism of in situ bacterial assemblages (Zubkov et al,
2001), and may therefore be uncoupled from the influence of ice on phytoplankton
activity. It is interesting to note, however, that several sharp increases in DMS occurred
simultaneously with the occurrence of small amounts of sea ice (<20% total cover) (Fig.
2, kms 3400 and 7200 on the cruise track). Limited station data also indicate high
DMS/P:chl *a* ratios in areas with a comparatively high sea ice cover, at stations BB3 and
CAA6 (Table 2).  At the time of our sampling, both of these stations were characterized
by very low phytoplankton biomass (0.11 μgL$^{-1}$ and 0.20 μg L$^{-1}$chl *a*, respectively) and
had particularly high DMSP: chl *a*  ratios (129 nmol μg$^{-1}$ and 182 nmol μg$^{-1}$,
respectively). This suggests a potential role for ice-edge effects, either through the melt-
induced stimulation of reduced sulfur production in DMS/P rich phytoplankton taxa, or
through the release of ice-associated DMS/P into the water column. Figures 2d and 2e
show decreased salinity in partially ice-covered areas, in particular around kms 4400,
7300, and 9200. Similar trends have been reported by several groups.  For instance,
Matrai and Vernet (1997)  reported significantly higher values of DMS and DMSP in
partially ice-covered waters of the Barents Sea relative to ice-free regions, while Gali et
al. (Gali et al., 2010) and Leck and Persson (1997) reported highest DMS/P values along
the ice edge in their Arctic surveys.
**4.6 DMS in a changing Arctic**




The Arctic marine ecosystem is currently undergoing a dramatic warming that is
expected to have far-reaching impact on phytoplankton dynamics and, likely, DMS
production and sea-air fluxes. Much of the ecosystem change is driven by warming and
rapidly melting sea ice, which influences mixed layer stratification, light regimes and
nutrient supply. Current work suggests that sea ice loss will eventually lead to a nutrient-
poor, shallow-stratified Arctic Ocean with low phytoplankton biomass (Levasseur, 2013).
Nutrient limitation may favor smaller cells, shifting diatom-dominated assemblages to
communities with a strong flagellate presence, and this may, in turn, increase DMSP
production and DMS emissions. A modeling study by Gabric et al (2005) projected
significant increases in DMS emissions in response to MLD shoaling and ice ablation.
Our observations from regions with shallow mixed layer depths and mixed phytoplankton
assemblages do indeed exhibit elevated DMSP:chl  ratios, providing some support for
this prediction.   On-going monitoring work will be needed to examine climate-driven
shifts in surface water productivity and biogeochemical cycles in Arctic Ocean waters.
**5. Conclusion**
We present a high spatial resolution dataset of reduced sulfur measurements
through the Canadian sector of the Arctic Ocean and Subarctic Atlantic.  We demonstrate
the utility of high-resolution DMS measurements for comparison with other
oceanographic variables, and show the coherence of DMS gradients with fine-scale
surface hydrographic structure, suggesting elevated DMS production in some
oceanographic frontal zones.  We also observed elevated DMS/P values in partially ice-
covered regions, suggesting that ice-edge effects may stimulate DMS/P production. Our

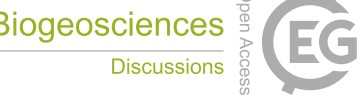

data serve to significantly expand the existing spatial coverage of reduced sulfur
measurements in the Arctic, providing a baseline for future studies in this rapidly
changing marine environment.  Future warming of surface waters and sea-ice melt could
lead to increased concentrations and sea-air fluxes of DMS, though significantly more
observations will be needed to substantiate this.
**Acknowledgements:** This work was supported by grants from the Natural Sciences and
Engineering Research Council of Canada (NSERC) through the Climate Change and
Atmospheric Research program (Arctic-GEOTRACES). We are grateful to the captain
and crew of the *CCGS Amundsen* for their invaluable support in this work.

**Data Availability:**
All data are available at the following github repository:
**https://github.com/tjarnikova/Jarnikova_Canadian_Arctic_DMS_supldata** (**DOI:**
**10.5281/zenodo.160225**)



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



**Tables:**

| Pigment | Associated Taxa |
|---|---|
| **Chlorophyll c$_3$** | Haptophytes |
| **Peridinin** | Dinoflagellates |
| **19'-butanoyloxyfucoxanthin** | Haptophyte |
| **Fucoxanthin** | Diatoms, Haptophytes |
| **19'-hexanoyloxyfucoxanthin** | Haptophytes, Dinoflagellates |
| **Diadinoxanthin** | Haptophytes, Dinoflagellates, Diatoms |
| **Violaxanthin** | Dinoflagellates |
| **Zeaxanthin** | Dinoflagellates |


**Table 1.** HPLC marker pigments and their associated phytoplankton taxa. Adapted from
(Coupel et al. 2015) .











| Station | Lat(N) | Lon(E) | MLD(m) | % Ice Cover | chla (ug L$^{-1}$) | DMS/chla (nmol µg$^{-1}$) | DMSP/chla (nmol µg$^{-1}$) | Perid/chla | 19'ButFuc/chla | Fuc/chla | 19'HexFuc/chla | Diadino/chla |
|---|---|---|---|---|---|---|---|---|---|---|---|---|
| K1 | 56.12 | -53.37 | 18.4 | nd | 0.51 | 6.6 | nd | 0.043 | 0.077 | 0.184 | 0.156 | 0.056 |
| LS2 | 60.45 | -56.55 | 41.4 | nd | 0.59 | 3.4 | nd | 0.051 | 0.012 | 0.277 | 0.025 | 0.024 |
| BB3 | 71.41 | -68.59 | 8.2 | 19.7 | 0.12 | bdl | 129.4 | 0.049 | 0.011 | 0.278 | 0.051 | 0.087 |
| BB2 | 72.75 | -67.00 | 10.3 | nd | 0.19 | 21.7 | 93.3 | 0.050 | 0.015 | 0.312 | 0.089 | 0.072 |
| CAA1 | 74.52 | -80.56 | 32.1 | nd | 0.56 | 6.9 | 52.3 | 0.015 | 0.018 | 0.239 | 0.023 | 0.042 |
| CAA5 | 74.12 | -91.49 | 5.3 | 6.61 | 0.16 | bdl | 114.7 | 0.078 | 0.017 | 0.326 | 0.020 | 0.051 |
| CAA6 | 74.75 | -97.47 | 6.1 | 16.43 | 0.21 | 10.6 | 181.7 | 0.054 | 0.021 | 0.401 | 0.015 | 0.058 |
| CAA7 | 73.66 | -96.53 | 2.1 | 13.3 | 0.13 | 15.6 | 81.3 | 0.109 | 0.066 | 0.335 | 0.057 | 0.146 |
| VS | 69.16 | -100.69 | 8.4 | 8.23 | 0.18 | 10.6 | nd | 0.029 | 0.020 | 0.309 | 0.032 | 0.037 |

**Table 2.** Mixed layer depth (MLD), ice cover, HPLC pigment measurements (ratios of selected marker pigments to chl a), DMS (MIMS) and DMSP (OSSCAR) measurements. Perid = peridinin, 19'ButFuc = 19'-butanoyloxyfucoxanthin, Fuc = Fucoxanthin, 19'HexFuc = 19'-hexanoyloxyfucoxanthin, Diadino = Diadinoxanthin *nd*= no data. *bdl* = below detection limit.





| Variable | DMS Correlation Coefficient | DMSP Correlation Coefficient |
| --- | --- | --- |
| $\Delta O_2/Ar$ | 0.22 | 0.33 |
| Salinity | 0.35 | 0.34 |
| SST | 0.29 | 0.14 |
| Fluorescence | 0.32 | 0.66 |
| $pCO_2$ | 0.16 | 0.12 |
| Ice Cover | -0.26 | -0.34 |


**Table 3** Pearson correlation coefficients relating DMS measurements made by MIMS and
DMSP measurements made by OSSCAR to other oceanographic variables. Only
correlations significant at the $p< 0.05$ level are shown.  $\Delta O_2/Ar$ ratios were obtained using
MIMS.












| Station | Diatom | Dinoflag. | Chloro. | Prasino | Crypto. | C-P | c3-Flag. | Hapto-7 |
|---------|--------|-----------|---------|---------|---------|-----|----------|---------|
| K1   | 37 | 14 | 0  | 17 | 4 | 9 | 1  | 16 |
| LS2  | 39 | 19 | 0  | 23 | 1 | 3 | 7  | 8  |
| BB3  | 48 | 15 | 4  | 14 | 8 | 1 | 5  | 5  |
| BB2  | 44 | 16 | 11 | 14 | 4 | 2 | 1  | 8  |
| CAA1 | 47 | 4  | 0  | 39 | 2 | 2 | 4  | 2  |
| CAA5 | 50 | 19 | 1  | 10 | 3 | 2 | 14 | 1  |
| CAA6 | 52 | 16 | 1  | 8  | 3 | 2 | 17 | 1  |
| CAA7 | 46 | 11 | 4  | 17 | 8 | 8 | 0  | 5  |
| VS   | 67 | 8  | 0  | 11 | 3 | 3 | 6  | 3  |


**Table 4** CHEMTAX-derived phytoplankton assemblage estimates (numbers given are
percent of total chl a) for sampled stations. Diat. = diatoms; Dinoflag = Dinoflagellates;
Chloro. = Chlorophytes; Prasino = Prasinophyte (types 2 and 3); Crypto. = Cryptophytes
Chryso-Pelago =Chrysophytes/Pelagophytes; c3-flag. = c3-Flagellates; Hapto-7 =
Haptophyte type 7. Due to the presence of unidentified phytoplankton taxa, not all
assemblage estimates sum to 100%.













| Author | Year | Month | Region | DMS (nM) | DMSP (nM) | Assemblage characteristics |
|---|---|---|---|---|---|---|
| **Bouillon et al. (2002)** | 1998 | April-June | North Water | 0.04-6.7 | 0.9-53 | Diatom dominated assemblage |
| **Matrai et al. (1997)** | 1993 | May | Barents Sea | 2.8 - 25.3 | 6-27 | Diatom-dominated and *Phaeocystis*-dominated stations |
| **Lizotte et al. (2012)** | 2003 | May-October | Northwest Atlantic | 0.1-12 | 4-101 | Nanoflagellate dominated in all seasons |
| **Gali et al. (2010)** | 2007 | July | Greenland Sea | 0.1 - 18.3 | 1.4 - 163.6 | Haptophyte (*Phaeocystis*) dominance |
| **Leck et al. (1996)** | 1991 | August-October | Greenland Sea | 0.04 - 12 | -- | Not described |
| **Motard-Côté et al. (2012)** | 2008 | September | Baffin Bay North Water | 0.4-5.2 | 5-70 | |
| **Scarratt et al. (2007)** | 1999 | September | Northwest Atlantic | 0.2-4.7 | 0-203 | Mixed assemblage |
| **Luce et al. (2011)** | 2007 | October-November | High Arctic | 0.05-0.8 | 2-39 | Flagellate-dominated except for diatom-dominated in Baffin Bay |
| **This study** | 2015 | July-August | Canadian Arctic Archipelago | 0.1-18 | <1 - 160 | Mixed assemblage, diatom-dominated |

**Table 5.** Compilation of published Arctic and Subarctic Atlantic DMS/P data from the
summer and fall months, focusing on observations from the Western Hemisphere.



**Figures:**

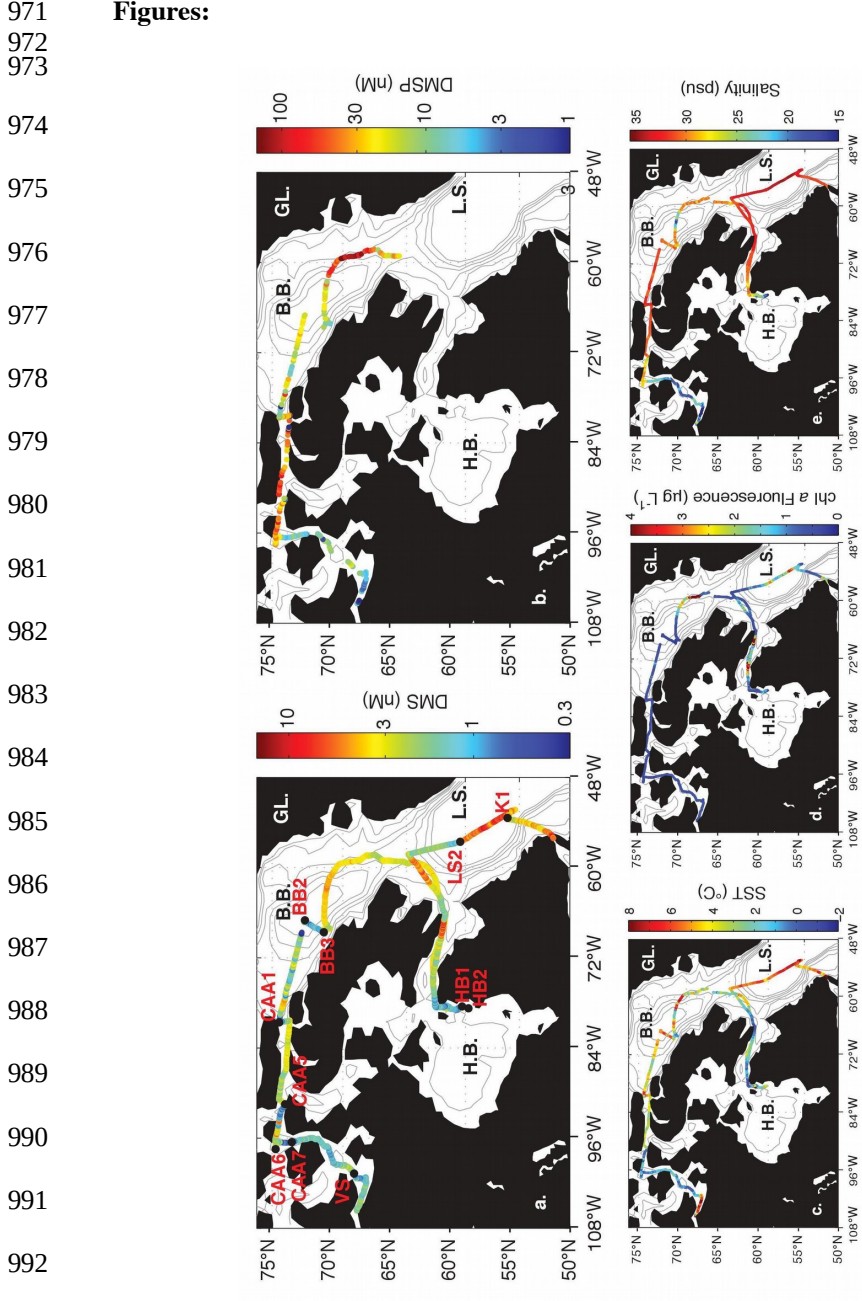

**Figure 1.** Spatial distribution of DMS, DMSP and hydrographic variables. GD. =
Greenland, B.B. = Baffin Bay, L.S = Labrador Sea, H.B. = Hudson Bay.





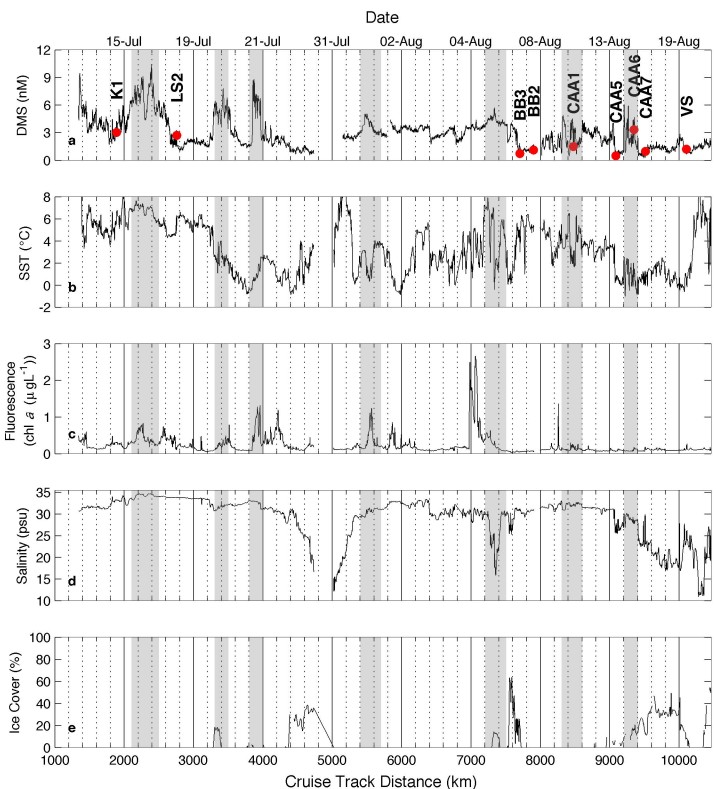

**Figure 2.** Distribution of DMS and hydrographic variables along our cruise track. Grey
shaded areas show denote regions of sharp increases in DMS.















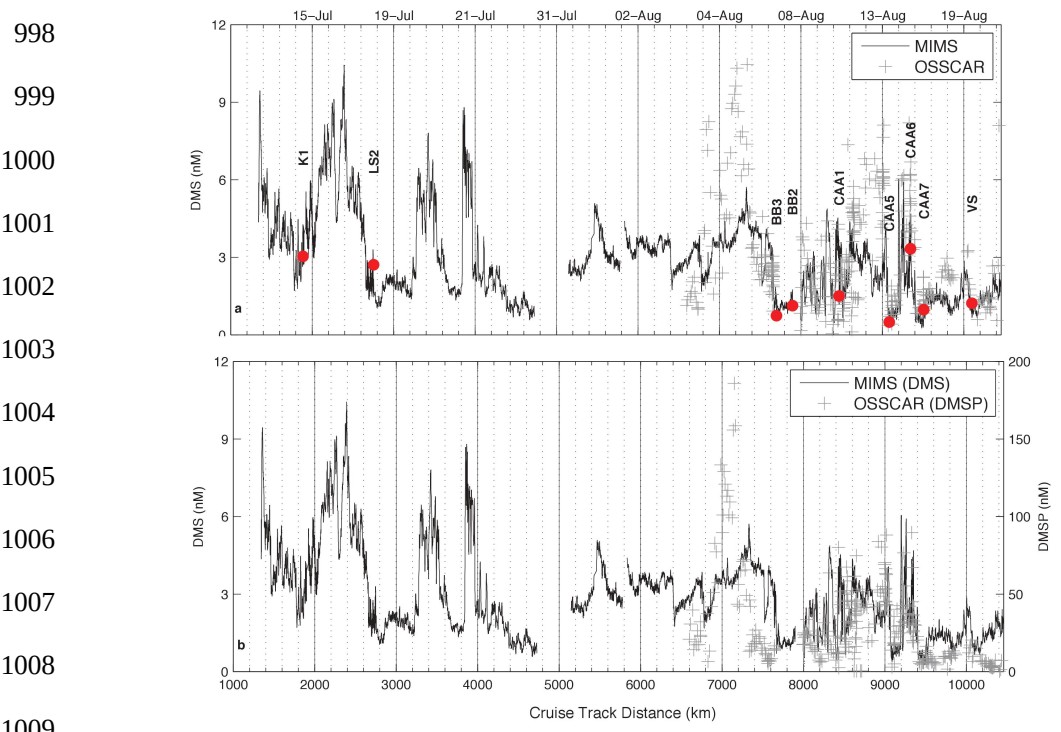

**Figure 3.** Distribution of DMS and DMSP along the cruise track. Panel (a) shows DMS
measurements made by MIMS and OSSCAR. Note that a small fraction (less than
0.5%) of measurements made by OSSCAR were above 12 nM. Panel (b) shows MIMS
data with OSSCAR DMSP measurements superimposed on a different y scale (right hand
side).



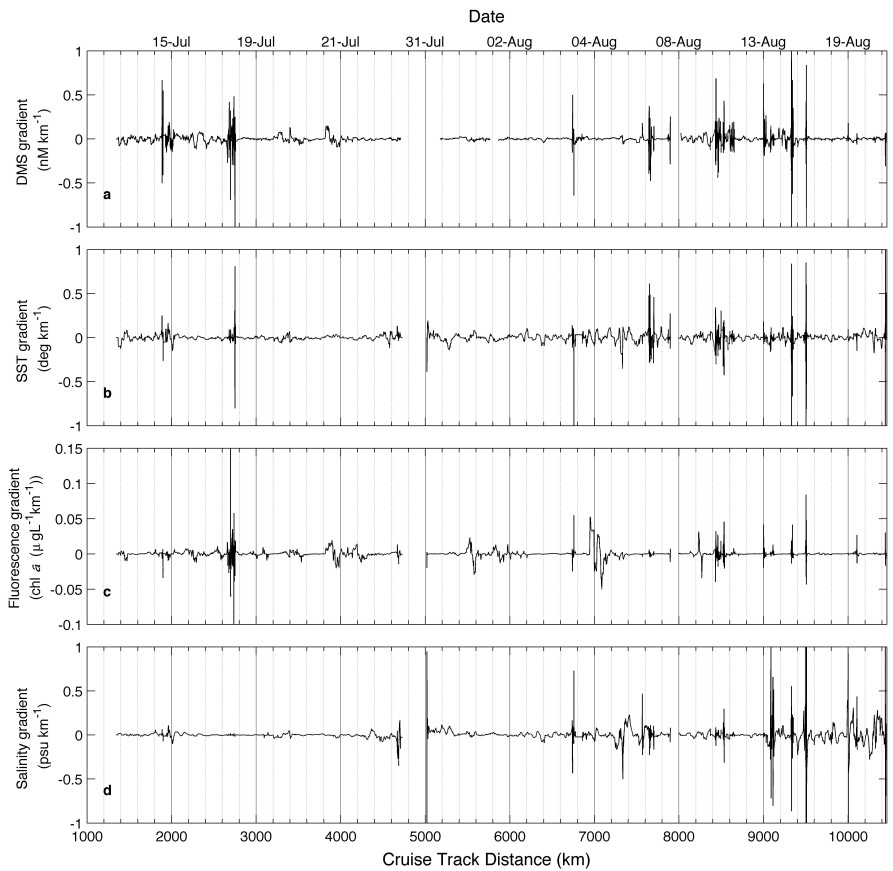

**Figure 4.** Spatial gradients in DMS and hydrographic variables.

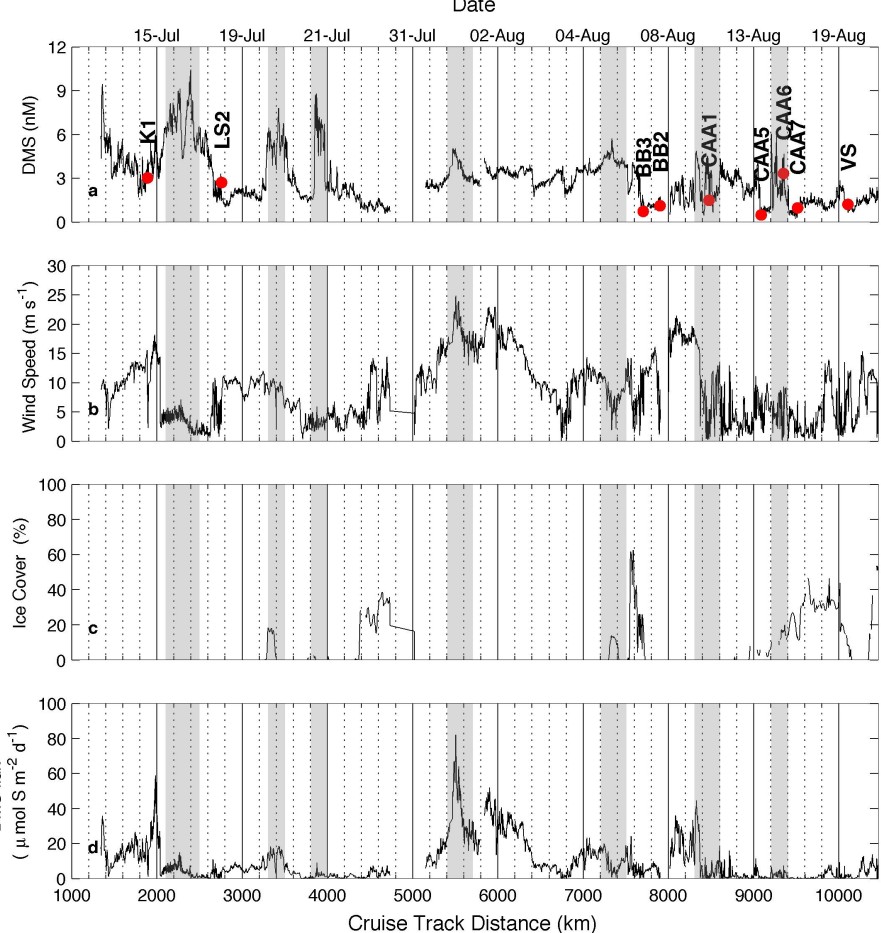






**Figure 5.** Distribution of DMS, wind speed, sea ice cover and sea-air DMS flux along the
cruise track.




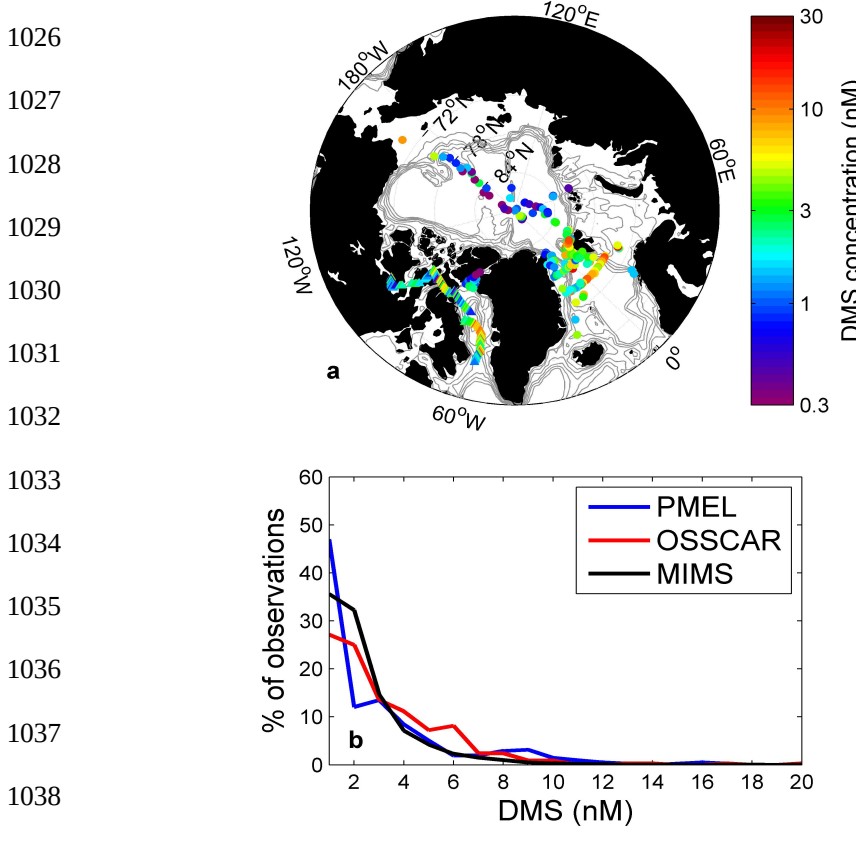

**Figure 6.** Comparison of OSSCAR- and MIMS-measured DMS from this study with
existing data in the PMEL database