# Peer review of "Tereza Jarníková (1), Jo"

_Biogeosciences, 2017_

## Referee Comment (RC1) · Anonymous Referee #1 · 4 Oct 2017

This manuscript presents DMS/P data measured in Canadian waters using two techniques, a MIMS and an automated GC-PFPD. The authors were able to use the fine resolution spatial distribution of sulfur compounds measured to examine the influence of frontal features and other small scale hydrographic changes on DMS/P. The authors provide a comprehensive introduction to DMS/P cycling and their importance in both the ocean and the atmosphere. They note that high latitude DMS emissions may be especially important for aerosol formation and polar climate. However, the number of measurements in these high latitude regions is scarce, compared to the mid- and low latitudes. The difference between findings in the Antarctic (high values of sulfur compounds) vs. the Arctic (moderate level of sulfur compounds) motivated this study

**BGD**

and the authors are particularly interested in the role Arctic sea ice plays on regulating DMS/P distributions. This manuscript is an important contribution to the DMS/P database and should be published after the following minor changes have been made.

Specific comments:

Lines 119-120: Is this Gabric reference the most updated reference on the feedback between ice albedo and DMS emissions?

Lines 199-216: What is the LOD for the MIMS?

Line 211: Perm tubes are highly sensitive to constant temperature and flow conditions. How reliable are these as primary standards when taken to sea?

Line 264: Why are your fluxes computed with N00, when more evidence is coming online that DMS k values should be linearly dependent on wind speed?

Lines 269-270: What your wind speed corrected to 10 m height?

Line 291: Do the authors mean Table 3 here instead of Table 4?

Lines 305-306: The measured range reported is way below the LOD. The authors discuss this much later, but maybe here there should be a statement about 22% of these are below the LOD.

Line 329: Do the authors mean Figure 4 here instead of Figure 5?

Line 379: Typo, remove of

Lines 410s: Are there no possible scenarios in which the MIMS values are too low? E.g. peak resolution not achieved because MIMS is too slow?

Lines 449-451: The top figure in this graph would be more instructive if we could see the comparison between this study and previous studies. The bottom figure helps with this, but does not give an idea of the spatial comparison.

Lines 455-rest of paragraph: Why is there no comparison to the Lana climatology here?

Section 4.3: There is only one reference here (Tremblay et al., 2011) related to DMS/P and fronts. Are there no others to corroborate the authors' findings?

Lines 533-535: There are no obvious trends in the data between MLD and sulfur compound concentrations. I am not sure that the following explanation is justified by the data.

Line 537: There appears to be something wrong with the numbers here. The shallowest MLD is 2.1 m in Table 2.

Lines 552-563: Are there no possible other explanations beside PFTS? Was there more bacterial activity? Or more cell lysis?

Lines 565-566: Are there no citations for this sentence? Is this considered common knowledge?

Line 576: Typo, extra space between study and comma

Line 585: What is 30a? Is this a citation typo?

Lines 590-592: In Table 2, I can see the highest sulfur:chl for stations BB2 and CAA7 for DMS. BB3 and CAA6 are for DMSP only.

Figure 1: Caption – GD should be GL

Figure 2: No description of red dots.

---

## Referee Comment (RC2) · Anonymous Referee #2 · 15 Oct 2017

General comments.

The study reports high spatial resolution measurements along a cruise track that passes through a number of distinct regions around the western Arctic. This is interesting on two counts, one is the high spatial resolution of the data that illustrates spatial gradients generally not observed using other approaches, and the second is the contribution to the comparatively few measurements of DMS/P that have been carried out in the Arctic in general and particularly in this region. These high resolution seawater measurements of DMS and DMSP are generated using MIMS and an OSS-CAR system that is probably unique to this group and the two systems have seldom

been applied simultaneously (e.g. Asher et al. 2015). This is an important data set and may well be useful to those trying to model DMS emissions in Arctic waters and the role that DMS may play in aerosol and cloud formation over the Arctic. Despite the uniqueness and quality of this data, in general, I think the authors fail to make full use of the high spatial resolution data and supporting information. For instance, much of the manuscript, including 3 tables, is dedicated to trying to identify the phytoplankton sources of DMSP and DMS from a limited dataset (9 stations along a 10,000 km transect) of pigment concentrations. It would be much more informative in my view, to concentrate on the unique high resolution data over the very long transect; especially what may be causing the large gradients neatly illustrated in Figure 4 and whether there are areas of particularly high or low DMS sea to air flux. I think the Discussion in particular needs to be more focused on the results from this dataset and what they might mean to DMS emissions in the regions.

Specific Comments

Abstract.

This could be tightened-up so that it really represents the finding in the main manuscript. At the moment it does little to convey the real relevance of the project.

L21-22. What does the conclusion that a range in concentrations of DMS (∼1 nM to 18nM) and DMSP concentrations (∼1 nM to150 nM) was consistent with previous observations in the Arctic Ocean really mean? This would apply to almost any large stretch of ocean wouldn't it?

L23. The comment about Baffin Bay is interesting but I do not see a focus on it in the actual manuscript, maybe there should be?

Introduction.

L41. The uncertainty in the CLAW hypothesis should also be made clear.

L48. Stefels et al. 2007 is now 10 years old, it might be worth considering whether

more recent studies have thrown new light on the topics?

L60. I don't think Zubkov et al. 2001 directly addresses stimulation of DMS production by grazing or viral lysis.

L68. Several modelling studies also suggest a limited role for DMS in cloud formation in the Arctic and should be mentioned (e.g. Carslaw et al 2012, Browse et al 2014 ).

L80 It is not clear what the relevance of this comparison between Arctic and Antarctic measured DMS values is, both datasets are regionally and seasonally biased making it difficult to conclude anything from the comparison of the full datasets.

L89-102. The relevance of the comparison of Arctic and Antarctic DMS concentrations and controls on that production is not clear at this point. This is not a component of either the Results or Discussion. Maybe this comparison would be more interesting and relevant as an aspect of the Discussion?

Methods

L159+ It would be useful to know why the data from the OSSCAR system does not cover the full transect, maybe I have missed that in the manuscript?

L181. It would be useful to include the concentration of the point standard as this would provide context for the standard error of 0.55 nM that is deemed the level of precision. Was this not concentration dependent?

L266. Flux estimates: possibly understandably the authors use a fairly simplistic parameterization to compute DMS exchange rates, but it should be noted that the Nightingale 2000 parameterization has now consistently been shown to overestimate flux at higher wind speeds. At some point we as a community are going to have to start using a more realistic parameterization. Plus the scaling exponent (0.4) derived from Loose et al. 2009, requires more explanation. Does this account for flux through the ice or for enhanced exchange due to turbulence generated by the ice etc.? A short section, possibly in the Discussion, is required to make this uncertainty clear.

L269. Was the wind corrected to U10 as is generally used in the Nightingale 2000 parameterization and was it corrected for ship speed?

Results

L364. It would be useful to have an indication of what distance the subjective 100 points refers to over which the gradients are calculated.

Figures. In general, the figure legends could be made more informative.

Discussion.

L406. Could this be caused by carryover of NaOH from DMSP analysis to DMS analysis? High concentrations of NaOH are difficult to wash off with only MQ water, was this tested with DMSP standards at all, i.e. purging of a DMSP standard following a DMSP analysis with NaOH addition.

L451. This comparison of DMS flux does not 'prove' anything really without access to the modeled information.

L462. Do you mean 'sulfur accumulation', or what does sulfur accumulation mean?

L477. It's not clear what point is being made here, I think the logic may be reversed?

L481. Again, it is not clear what point is being made in this paragraph and whether it is necessary?

L492. The Simo and Pedros-Alio (1999) study was based on experiments in the North Atlantic, so also not restricted to lower latitudes, and many studies have confirmed the 'summer paradox' seasonal pattern extends beyond low latitude waters.

L497. I think this section would be made more interesting if it was considered in terms of trying to develop a seasonal model that included DMS emissions. Is there sufficient information (Table 5 and this study) to start to develop such a model? If not, what is needed; seasonal studies in different regions of the Arctic or more transects throughout

the year?

L502.' decorrelation length scales' needs more explanation, especially as the authors then go on to point out co-occurring gradients.

L510. I'm not sure whether the argument is consistent - why should high primary production drive increased DMS - as pointed out earlier in the text, high nutrient periods on a seasonal basis are associated with low DMS.

L523. I'm not sure where this paragraph goes, other than to highlight a different study by these authors? Are there regions identified in this region of the Arctic where understanding the processes would be particularly useful and how might that be achieved?

L532/Section 4. This section starts off discussing areas of shallow mixed layer depth then merges into a discussion comparing DMSP:chl a ratios generally in the Arctic. What point(s) is being conveyed?

L552. Again, it is unclear what point is being made here - the range in DMSP:chl is > 3-fold, (52 - 182), why does the final sentence conclude low variability in DMSP:chl?

L564. This section could be usefully focused. It would be more relevant to focus on what this dataset shows. The paragraph from L565 simply reviews previous studies and seems superfluous. As it stands, it is unclear what the authors conclude. The correlation between sea ice cover and DMS/P is negative but station-specific data suggests enhanced DMSP:chl ratios near the ice edge?

L558. Does the MIMS data pick-up ice edge effects on DMS concentration, i.e. is DMS related to salinity when passing through marginal ice zones or ice edges, over shorter distances than the whole dataset?

L605 Again this section reads very much like a review, with little focus on what this particular study demonstrates.

Minor points

L144 - use CAA instead of Canadian. . .

L196 –should read 'convert DMSO to DMS'

L266, more correctly A is proportion of sea ice cover, rather than percentage.

Figure 1. GL - Greenland not GD; and please define CAA

Figure 2.please explain what the red dots are on the DMS graph, possibly station mesurements? This would strengthen the suggestion made on L595+

Figure 3. please explain what the red dots are?

Figure 6. Some more information in the legend would be useful, for instance, how are the data compiled, what exactly is illustrated? The total number of points for each dataset would also be useful.

Please also note the supplement to this comment:
https://www.biogeosciences-discuss.net/bg-2017-337/bg-2017-337-RC2-supplement.pdf

---

## Referee Comment (RC3) · Anonymous Referee #3 · 1 Nov 2017

1. General Comments In this paper, authors measured the concentrations of DMS and DMSP in the surface seawater from the Labrador Sea to the Canadian Arctic Archipelago during the summer of 2015. In addition to the distributions of DMS/P concentrations with information of seawater parameters (hydrographic parameters, taxonomic compositions, and sea ice cover) in this area, they detected the abrupt increases in the DMS concentration at the front of hydrographic parameters by measurement with fine spatial resolution. The data obtained from this observation contributes to the accumulation of database in the Canadian Arctic waters and to discussion on the response of DMS to changes occurring in the Arctic Ocean. This paper would be acceptable if the authors reconsider and correct the part described in Specific Comments and

[Figure]

Technical Corrections.

2. Specific Comments

(1) Authors conclude that the results obtained from this cruise (shallow MLDs and mixed phytoplankton assemblages) support the results of previous studies by Gabric et al. (2004) and Levasseur (2013). I wonder whether the spatial changes such as the differences between the sea-ice free area and the locations where sea ice has just melted can be compared with the result caused by sea ice reduction in future Arctic waters.

(2) The authors discuss the sharp increase in DMS concentration obtained from the measurement of high spatial resolution. Then what happened where the DMS concentration sharply decreased for example at 2600km, 3600km, 4000km, and 7600km?

(3) L262: In Equation (1), do you need to include the DMS concentration in the surface atmosphere? When you use this equation, you need to mention about omitting this.

(4) L269: The exchange coefficient of the flux calculation is a function of the wind speed at the 10m height from the sea surface. Please note that the height of the anemometer. And if the height is far from 10m, then you need to mention about the influence of the height difference.

3. Technical Corrections

(1) When reading bg-2017-337.pdf, I found the following typing mistakes. I think that there are other typing mistakes in this draft, so reread and fix them.

(2) About the use of "Figure" and "Fig", if "Figure" is used at the beginning of a sentence and "Fig." is used in sentences, "Figure 3" at L317 is not at the beginning of the sentence, is this correct?

(3) Figure 5 is referred (L329) prior to Figure 4 (L368). You need to swap Figure 4 and Figure 5.

(4) Table 4 is referred prior to Table 3. You need to swap Table 3 (L375) and Table 4 (L291). The order of Table 3 and Table 4 is also the same

(5) Is the writing style "km 7000" in L309 correct? This notation can be found in several other places such as L309, L322, L326, L331, L334, L336, L343, L346, L386, L403, L590, and L598.

(6) In "Reference list", journal names written by full notation and abbreviations are mixed.

(7) In Figure 2, gray shaded areas denote not only the part of sharp increase in DMS concentration, but also the part of its high concentration (and its sharp decrease). If you want to highlight only the part of its sharp increase, carefully mark the part only where the DMS concentration increases.

(8) List of technical errors L53: Is "three order" correct?

L59: At the end of L59, not period but comma.

L70~L71: Invalid way to cite the references written in L70-L71; DMS emissions (Chang et al., 2011), (Mungall et al., 2016) ==> DMS emissions (Chang et al., 2011; Mungall et al., 2016)

L92: mixed layer depth => mixed layer depth (MLD) L234, L280, L290, L534, L615, L914: mixed layer depth => MLD

L128, L155, L200: The notation of "membrane inlet mass spectrometry (MIMS)" is to be used only at the first quotation, and the abbreviation (MIMS) should be used at subsequent citations.

L130, L156 The notation of "OSSCAR" is same as my comment for "MIMS".

L138: "(July 10-August 20, 2015)". Is parenthesis "( )" necessary?

L147: shallow, narrow straits => shallow and narrow straits

L263: Where => where

L306: 18nM => 18 nM

L316, L317: 29nM => 29 nM, insert half-size space after "29" 52.31mmol => 52.31 nmol

L314: Is "(measured with OSSCAR)" necessary? If so, ( ) is necessary?

L331: sea-air flux observed => sea-air flux calculated (or estimated)

L348: (Fig 2c) => (Fig.2c) insert period after "Fig".

L360: Is (G) in "Gradients (G) for each variable" necessary here?

L416: Quotation "Wolfe et al (2002)" at the end of L416 and the same author's quotation at the end of the sentence of L418 are duplicated (the latter is unnecessary).

L444: Fig 6a,=> Fig. 6a

L446: Insert a half-size space between "(Fig. 6b)." and " The result,,,,".

L459: heavily => mainly

L461: drawn => obtained

L474: in the Northwest Subarctic => in the northwest subarctic

L474: Lizotte et al (2012) => Lizotte et al. (2012)

L500: Either "Previous" in L500 or "previously" in L501 is unnecessary.

L514: (Tremblay et al., 2011) in the end of sentence should be deleted.

L540: (Matrai et al. 1997) => (Matrai et al., 1997) Need comma after "et al."

L553, L554, L556: nM ug-1 => nmol ug-1

L560ïïjŽDMSP:Chl and HPLC => insert "a" after chl in italic. DMSP:Chl a and HPLC

L585: [30a] Reference?

L601-L602: Gali et al. (Gali et al., 2010) => Gali et al. (2010)

L613: Gabric et al (2005): Need period after "et al". Year of publish of this paper is 2004, not 2005.

L616: DMSP:Chl a ratios :Chl a should be italic.

L844: 20244. => 20244 (Delete period after 20244.)

L885: Adapted from (Coupel et al. 2015). => Adapted from Coupel et al. (2015).

L997: show denote =>Choose either "show" or "denote".

L1024-L1025: What are the grey area in these figures? Are the same as Fig.2?

L1040-L1041: Explain the panel (a) and (b).

---

## Author Comment (AC1) · 11 Dec 2017

Dear Anonymous Referee #1,

Thank you for your thoughtful and constructive criticism of our work. We have reread and revised our manuscript according to the insightful suggestions you provided, and have added some key references. The most substantial revision may consist of changing the DMS flux calculation used, and of correcting our wind speed to 10m height. This is not a problem, as the data are readily available. Please find responses to each of your points, below. We have done our best to address each statement carefully, and look forward to your responses. An updated manuscript is also available.

[Figure]

Best, Tereza Jarnikova PhD Candidate, UBC
Specific comments: Lines 119-120: Is this Gabric reference the most updated reference on the feedback between ice albedo and DMS emissions?

>We have added a reference to a quite recent modelling study by Cameron-Smith et al, run for the Southern Ocean, that also demonstrates a remarkably strong DMS emission response to loss of sea-ice albedo. Becagli et al (2016) (http://dx.doi.org/10.1016/j.atmosenv.2016.04.002) also observed a robust correlation between DMS-sourced aerosol concentration and sea-ice melt, though we did not add this citation as the section focuses on modelling studies of future polar regions.

Lines 199-216: What is the LOD for the MIMS?

>A line has been added here - 2nM, reference Tortell 2005.

Line 211: Perm tubes are highly sensitive to constant temperature and flow conditions. How reliable are these as primary standards when taken to sea?

>Both these sensitivities are thoroughly addressed - the temperature of the perm tube is kept constant by use of a circuit-controlled heating pad, and the flow through it is kept constant via a flow gauge. We now explicitly state this in our manuscript.

Line 264: Why are your fluxes computed with N00, when more evidence is coming online that DMS k values should be linearly dependent on wind speed?

>We used this flux computation because it is consistent with the one used by the main global DMS climatology, Lana et al. However, we are aware that newer computations have been published, eg. Bell et al. It would be possible to use these.

Lines 269-270: What your wind speed corrected to 10 m height?

>This is an oversight on our part - the initial data were not corrected to 10m. However, our collaborators have provided us with this data, and we will revise this figure and the discussion as necessary.

Line 291: Do the authors mean Table 3 here instead of Table 4?

>Thank you! This has been fixed

Lines 305-306: The measured range reported is way below the LOD. The authors discuss this much later, but maybe here there should be a statement about 22% of these are below the LOD.

>For maximum clarity, a line has been added here briefly discussing this, and alluding to the more extensive discussion further down.

Line 329: Do the authors mean Figure 4 here instead of Figure 5?

>Thank you! This has been fixed

Line 379: Typo, remove of

>Thank you! This has been fixed

Lines 410s: Are there no possible scenarios in which the MIMS values are too low? E.g. peak resolution not achieved because MIMS is too slow?

>We are not aware of any reason that the MIMS should under-estimate DMS. We use a ∼30 second dwell time to ensure good peak resolution for DMS at m/z 62.

Lines 449-451: The top figure in this graph would be more instructive if we could see the comparison between this study and previous studies. The bottom figure helps with this, but does not give an idea of the spatial comparison. >The top panel in this figure does show a comparison between the current study and previous studies both in terms of concentration distribution and spatial distribution. We have changed the symbols to help clarify the presentation of different data sources.

Lines 455-rest of paragraph: Why is there no comparison to the Lana climatology here?

>The Lana climatology is based on the PMEL measurements, which we compare with directly. Any information present in the Lana climatology that is not present in the PMEL data results simply from interpolations, and thus (we feel) are not worth comparing. There is almost no data in the PMEL database in our observational region (the Canadian Arctic) - almost all PMEL measurements in the Boreal Polar Longhurst province come from the Atlantic sector. Therefore, according to Lana's methodology, the Lana climatology presents only a rough "first guess" of concentrations in the studied region.

Section 4.3: There is only one reference here (Tremblay et al., 2011) related to DMS/P and fronts. Are there no others to corroborate the authors' findings?

>Here we are focusing on the idea that frontal zones may be regions of enhanced productivity, and we then argue that this may lead to increased DMS production. This idea is well-established in the literature - eg Lutjeharms 1985 and others. We chose the Tremblay reference because it is from a similar region from a similar time of year.

[Figure]

Lines 533-535: There are no obvious trends in the data between MLD and sulfur compound concentrations. I am not sure that the following explanation is justified by the data.

>We have changed the wording here somewhat to reflect the lack of an overall statistical trend.

Line 537: There appears to be something wrong with the numbers here. The shallowest MLD is 2.1 m in Table 2.

>This was a typo; I have removed this sentence.

Lines 552-563: Are there no possible other explanations beside PFTS? Was there more bacterial activity? Or more cell lysis?

>We agree that these factors are important, though it is not possible for us to estimate them from our observational data. We have added a comment about these factors in this section.

Lines 565-566: Are there no citations for this sentence? Is this considered common knowledge?

>We now cite a review paper published by Levasseur in 2013, which gives a good overview of the relevant literature.

Line 576: Typo, extra space between study and comma

>Thank you, this has been fixed.

Line 585: What is 30a? Is this a citation typo?

>This has been corrected - this is a reference to the Galindo 2014 paper.

Lines 590-592: In Table 2, I can see the highest sulfur:chl for stations BB2 and CAA7 for DMS. BB3 and CAA6 are for DMSP only.

>We have changed this sentence to reflect DMSP only. In this way, the section makes

a better link between DMSP:chl ratios and sea ice cover.

Figure 1: Caption – GD should be GL

>Corrected – thank you.

Figure 2: No description of red dots.

>Corrected – thank you.

---

## Author Response (AR1)

This manuscript presents DMS/P data measured in Canadian waters using two techniques,
a MIMS and an automated GC-PFPD. The authors were able to use the fine
resolution spatial distribution of sulfur compounds measured to examine the influence
of frontal features and other small scale hydrographic changes on DMS/P. The authors
provide a comprehensive introduction to DMS/P cycling and their importance in both
the ocean and the atmosphere. They note that high latitude DMS emissions may be
especially important for aerosol formation and polar climate. However, the number
of measurements in these high latitude regions is scarce, compared to the mid- and
low latitudes. The difference between findings in the Antarctic (high values of sulfur
compounds) vs. the Arctic (moderate level of sulfur compounds) motivated this study
and the authors are particularly interested in the role Arctic sea ice plays on regulating
DMS/P distributions. This manuscript is an important contribution to the DMS/P
database and should be published after the following minor changes have been made.

Specific comments:
Lines 119-120: Is this Gabric reference the most updated reference on the feedback
between ice albedo and DMS emissions?

>We have added a reference to a quite recent modelling study by Cameron-Smith et al, run for the Southern Ocean, that also demonstrates a remarkably strong DMS emission response to loss of sea-ice albedo. Becagli et al (2016) (http://dx.doi.org/10.1016/j.atmosenv.2016.04.002) also observed a robust correlation between DMS-sourced aerosol concentration and sea-ice melt, though we did not add this citation as the section focuses on modelling studies of future polar regions.
Line number in Tracked Changes (TC) manuscript: 130

Lines 199-216: What is the LOD for the MIMS?

>A line has been added here - 2nM, reference Tortell 2005.

*Line number in TC manuscript: 232*

Line 211: Perm tubes are highly sensitive to constant temperature and flow conditions. How reliable are these as primary standards when taken to sea?

*>Both these sensitivities are thoroughly addressed - the temperature of the perm tube is kept constant by use of a circuit-controlled heating pad, and the flow through it is kept constant via a flow gauge. We now explicitly state this in our manuscript.*
*Line number in TC manuscript: 225-230*

Line 264: Why are your fluxes computed with N00, when more evidence is coming online that DMS k values should be linearly dependent on wind speed?

*>We used this flux computation because it is consistent with the one used by the main global DMS climatology, Lana et al, and still widely used. However, we are aware that newer computations have been published, eg. Bell et al. It would be possible to use these, though we have not done so at present.*

Lines 269-270: What your wind speed corrected to 10 m height?

*>This is an oversight on our part - the initial data were not corrected to 10m. However, we have made the correction in the present iteration. A supporting figure showing the slight difference in windspeed is added.*

*Line number in TC manuscript: 289*

Line 291: Do the authors mean Table 3 here instead of Table 4?

*>Thank you! This has been fixed*
*Line number in TC manuscript: 310*

Lines 305-306: The measured range reported is way below the LOD. The authors discuss this much later, but maybe here there should be a statement about 22% of these are below the LOD.

*>For maximum clarity, a line has been added here briefly discussing this, and alluding to the more extensive discussion further down.*
*Line number in TC manuscript: 325*

Line 329: Do the authors mean Figure 4 here instead of Figure 5?

*>Thank you! This has been fixed*
*Line number in TC manuscript: 349*

Line 379: Typo, remove of

*>Thank you! This has been fixed*

Lines 410s: Are there no possible scenarios in which the MIMS values are too low?
E.g. peak resolution not achieved because MIMS is too slow?

*>We are not aware of any reason that the MIMS should under-estimate DMS. We use a ~30
second dwell time to ensure good peak resolution for DMS at m/z 62.*

Lines 449-451: The top figure in this graph would be more instructive if we could see
the comparison between this study and previous studies. The bottom figure helps with
this, but does not give an idea of the spatial comparison.
*>The top panel in this figure does show a comparison between the current study and previous
studies both in terms of concentration distribution and spatial distribution. We have made the
symbols clearer to help clarify the presentation of different data sources.*
*Line number in TC manuscript: Figure 6*

Lines 455-rest of paragraph: Why is there no comparison to the Lana climatology here?

*>The Lana climatology is based on the PMEL measurements, which we compare with directly.
Any information present in the Lana climatology that is not present in the PMEL data results
simply from interpolations, and thus (we feel) are not worth comparing. There is almost no data
in the PMEL database in our observational region (the Canadian Arctic) - almost all PMEL
measurements in the Boreal Polar Longhurst province come from the Atlantic sector. Therefore,
according to Lana's methodology, the Lana climatology presents only a rough "first guess" of
concentrations in the studied region.*
*Line number in TC manuscript: 465*

Section 4.3: There is only one reference here (Tremblay et al., 2011) related to DMS/P
and fronts. Are there no others to corroborate the authors' findings?

*>Here we are focusing on the idea that frontal zones may be regions of enhanced productivity,
and we then argue that this may lead to increased DMS production. This idea is well-
established in the literature - eg Lutjeharms 1985 and others. We chose the Tremblay reference
because it is from a similar region from a similar time of year.*
*Line number in TC manuscript: 532*

Lines 533-535: There are no obvious trends in the data between MLD and sulfur compound
concentrations. I am not sure that the following explanation is justified by the
data.

*>We have changed the wording here somewhat to reflect the lack of an overall statistical trend.*
*Line number in TC manuscript: 565*

Line 537: There appears to be something wrong with the numbers here. The shallowest
MLD is 2.1 m in Table 2.

*>This was a typo; I have rewritten this sentence.*
*Line number in TC manuscript: 565*

Lines 552-563: Are there no possible other explanations beside PFTS? Was there more bacterial activity? Or more cell lysis?

*>We agree that these factors are important, though it is not possible for us to estimate them from our observational data. We have added a comment about these factors in this section.*
*Line number in TC manuscript: 598*

Lines 565-566: Are there no citations for this sentence? Is this considered common knowledge?

*>We now cite a review paper published by Levasseur in 2013, which gives a good overview of the relevant literature.*
*Line number in TC manuscript: 605*

Line 576: Typo, extra space between study and comma

*>Thank you, this has been fixed.*

Line 585: What is 30a? Is this a citation typo?

*>This has been corrected - this is a reference to the Galindo 2014 paper.*
*Line number in TC manuscript: 631*

Lines 590-592: In Table 2, I can see the highest sulfur:chl for stations BB2 and CAA7 for DMS. BB3 and CAA6 are for DMSP only.

*>We have changed this sentence to reflect DMSP only. In this way, the section makes a better link between DMSP:chl ratios and sea ice cover.*
*Line number in TC manuscript: 637*

Figure 1: Caption – GD should be GL

*>Corrected – thank you.*

Figure 2: No description of red dots.

*>Corrected – thank you.*

Dear Anonymous Referee #2,

Thank you for your thoughtful and constructive criticism of our work. We have reread and revised our manuscript according to the insightful suggestions you provided, and have added some key references. The most substantial revision may consist of a reworking (and slight shortening) of our Discussion section in response to your points - we aim to be more cohesive and clear when placing our work in the context of other studies performed in the Arctic. Please find responses to each of your points, below. We have done our best to address each statement carefully, and look forward to your responses. An updated manuscript is also available.

Best,
Tereza Jarnikova
PhD Student, UBC

Interactive comment on "The distribution of
methylated sulfur compounds, DMS and DMSP, in
Canadian Subarctic and Arctic marine waters
during summer, 2015" by Tereza Jarníková et al.
Anonymous Referee #2

General comments.
The study reports high spatial resolution measurements along a cruise track that
passes through a number of distinct regions around the western Arctic. This is interesting
on two counts, one is the high spatial resolution of the data that illustrates
spatial gradients generally not observed using other approaches, and the second is
the contribution to the comparatively few measurements of DMS/P that have been carried
out in the Arctic in general and particularly in this region. These high resolution
seawater measurements of DMS and DMSP are generated using MIMS and an OSSCAR
system that is probably unique to this group and the two systems have seldom
been applied simultaneously (e.g. Asher et al. 2015). This is an important data set
and may well be useful to those trying to model DMS emissions in Arctic waters and
the role that DMS may play in aerosol and cloud formation over the Arctic. Despite
the uniqueness and quality of this data, in general, I think the authors fail to make full
use of the high spatial resolution data and supporting information. For instance, much
of the manuscript, including 3 tables, is dedicated to trying to identify the phytoplankton
sources of DMSP and DMS from a limited dataset (9 stations along a 10,000 km
transect) of pigment concentrations. It would be much more informative in my view, to
concentrate on the unique high resolution data over the very long transect; especially
what may be causing the large gradients neatly illustrated in Figure 4 and whether
there are areas of particularly high or low DMS sea to air flux. I think the Discussion
in particular needs to be more focused on the results from this dataset and what they
might mean to DMS emissions in the regions.

Specific Comments
Abstract.
This could be tightened-up so that it really represents the finding in the main manuscript. At the moment it does little to convey the real relevance of the project.

>*We feel that the abstract quite clearly conveys the main results of the paper. However, we have revised it to better reflect the broader significance of the work.*

L21-22. What does the conclusion that a range in concentrations of DMS (~1 nM to 18nM) and DMSP concentrations (~1 nM to150 nM) was consistent with previous observations in the Arctic Ocean really mean? This would apply to almost any large stretch of ocean wouldn't it?

>*Thank you, we agree and have removed this part of the sentence.*
*Line number in tracked changes (TC) manuscript: 23*

L23. The comment about Baffin Bay is interesting but I do not see a focus on it in the actual manuscript, maybe there should be?

>*We agree that the Baffin Bay DMS concentrations are interesting, and wish to highlight them. We discuss the sharp increase in DMS Baffin Bay concentration from lines 347-357 (line 364 in track changes manuscript), and contrast them with the rest of the transect.*

Introduction.
L41. The uncertainty in the CLAW hypothesis should also be made clear.

>*We have added a reference to the Quinn and Bates paper, stating that this mechanism remains the subject of debate.*
*Line number in TC manuscript: 47*

L48. Stefels et al. 2007 is now 10 years old, it might be worth considering whether more recent studies have thrown new light on the topics?

>*Though we agree the Stefels paper is older, it remains one of the most comprehensive reviews of the DMS cycle, which makes it very suitable for a general reference such as the one called for here. We cite more recent papers when discussing specific aspects of our findings.*
*Line number in TC manuscript: 53*

L60. I don't think Zubkov et al. 2001 directly addresses stimulation of DMS production by grazing or viral lysis.

>*We switched this reference to Evans 2007, who directly compares the relative significance of grazing and viral lysis in DMS production in an E. huxleyi culture study.*
*Line number in TC manuscript: 65*

L68. Several modelling studies also suggest a limited role for DMS in cloud formation in the Arctic and should be mentioned (e.g. Carslaw et al 2012, Browse et al 2014 ).

*>We have added a line stating this, and have cited the Browse 2014 study, as well as citing Carslaw's 2013 paper that highlights the importance of quantifying uncertainty natural aerosol contribution to climate forcing.*
*Line number in TC manuscript: 73*

L80 It is not clear what the relevance of this comparison between Arctic and Antarctic measured DMS values is, both datasets are regionally and seasonally biased making it difficult to conclude anything from the comparison of the full datasets.

*>We respectfully disagree - we feel that it is important to point out the different controls on DMS production in the two polar regions, which otherwise share a number of physical characteristics, including seasonally varying sea ice cover and insolation. The difference between these two polar regions is critical to understanding potential drivers of DMS cycling in these regions, and this provides context for our work in the Arctic.*

L89-102. The relevance of the comparison of Arctic and Antarctic DMS concentrations and controls on that production is not clear at this point. This is not a component of either the Results or Discussion. Maybe this comparison would be more interesting and relevant as an aspect of the Discussion?

*>See above - we believe that a brief discussion of the different DMS dynamics of the two polar regions provides context for our work in the Arctic. It is a fact that there has been a hugely disproportionate effort towards understanding DMS/P dynamics in south polar regions. The relative lack of data from the northern polar regions is a main motivating factor for our work.*

Methods
L159+ It would be useful to know why the data from the OSSCAR system does not cover the full transect, maybe I have missed that in the manuscript?

*>We experienced technical difficulties with the OSSCAR system in the early part of the expedition. This is now explicitly mentioned in the manuscript.*
*Line number in TC manuscript: 211*

L181. It would be useful to include the concentration of the point standard as this would provide context for the standard error of 0.55 nM that is deemed the level of precision. Was this not concentration dependent?

*>We agree - this inline standard was 20 nM, and this has been noted in the text.*
*Line number in TC manuscript: 189*

L266. Flux estimates: possibly understandably the authors use a fairly simplistic parameterization to compute DMS exchange rates, but it should be noted that the Nightingale 2000 parameterization has now consistently been shown to overestimate flux at higher wind speeds. At some point we as a community are going to have to start using a more realistic parameterization. Plus the scaling exponent (0.4) derived from Loose et al. 2009, requires more explanation. Does this account for flux through the ice or for enhanced exchange due to turbulence generated by the ice etc.? A short section, possibly in the Discussion, is required to make this uncertainty clear.

>*We agree - this point about the Nightingale parameterization has been brought up by Reviewer 1 as well. We used it to be consistent with Lana et al and much of the literature. We have also added a brief comment about the Loose et al scaling, which is experimentally derived in a laboratory setting and does not account for turbulence generated by sea ice melt - we have now stated this.*
*Line number in TC manuscript: 284*

L269. Was the wind corrected to U10 as is generally used in the Nightingale 2000 parameterization and was it corrected for ship speed?

>*This was an oversight on our part - the original ship data we used was corrected for ship speed but not corrected to U10, but we have obtained U10 data from collaborators and have corrected it in the revised manuscript.*
*Line number in TC manuscript: 286*

Results
L364. It would be useful to have an indication of what distance the subjective 100 points refers to over which the gradients are calculated.

>*In our dataset, a radius of 100 points corresponds to a mean radius of around 25 km; this has been clarified in the revised text.*
*Line number in TC manuscript: 384*

Figures. In general, the figure legends could be made more informative.

>*We have added additional details to the legends for figures 1,2,5, and 6, but it wasn't clear what the reviewer was exactly looking for.*

Discussion.
L406. Could this be caused by carryover of NaOH from DMSP analysis to DMS analysis? High concentrations of NaOH are difficult to wash off with only MQ water, was this tested with DMSP standards at all, i.e. purging of a DMSP standard following a DMSP analysis with NaOH addition.

>*Prior to the cruise, we did test the thoroughness of our rinse cycle by testing DMSP standards with no NaOH added, following a DMSP analysis with NaOH addition, and these blanks were clean – a note about this has been added to the text. Nevertheless we want to state this possibility.*
*Line number in TC manuscript: 424*

L451. This comparison of DMS flux does not 'prove' anything really without access to the modeled information.

>*We agree and have removed this sentence.*
*Line number in TC manuscript: 471*

L462. Do you mean 'sulfur accumulation', or what does sulfur accumulation mean?

*>The statement 'DMS and DMSP concentrations' is more exact here. We have rephrased this.*
*Line number in TC manuscript: 483*

L477. It's not clear what point is being made here, I think the logic may be reversed?

*>The purpose of this section is to discuss seasonal patterns in phytoplankton biomass, productivity and DMS/P concentrations, as observed by fellow researchers in the region. It serves to provide a context for the results we obtained in our study. We are not clear which logic the reviewer sees as reversed.*
*Line number in TC manuscript: 497*

L481. Again, it is not clear what point is being made in this paragraph and whether it is necessary?

*>See above comment.*

L492. The Simo and Pedros-Alio (1999) study was based on experiments in the North Atlantic, so also not restricted to lower latitudes, and many studies have confirmed the 'summer paradox' seasonal pattern extends beyond low latitude waters.

*>We have rephrased this to reflect observations beyond the low latitudes.*
*Line number in TC manuscript: 512*

L497. I think this section would be made more interesting if it was considered in terms of trying to develop a seasonal model that included DMS emissions. Is there sufficient information (Table 5 and this study) to start to develop such a model? If not, what is needed; seasonal studies in different regions of the Arctic or more transects throughout the year?

*We believe that a seasonal model of DMS in the Arctic is being developed by our colleagues at University of Victoria, (Dr. Steiner's group). We have not focused on model construction in the Discussion as we feel that it is beyond the scope of the paper, but a comparison of seasonal studies in different regions of the Arctic would certainly be helpful in model construction.*

L502.' decorrelation length scales' needs more explanation, especially as the authors then go on to point out co-occurring gradients.

*>Decorrelation length scales provide information on the spatial scale of processes driving the majority of variability in DMS concentrations. We have added an explanatory line to the text. .*
*Line number in TC manuscript: 524*

L510. I'm not sure whether the argument is consistent - why should high primary production drive increased DMS - as pointed out earlier in the text, high nutrient periods on a seasonal basis are associated with low DMS.

*>We can appreciate the apparent contradiction with our earlier discussion of seasonal changes, and have rewritten this section.*
*Line number in TC manuscript: 535*

L523. I'm not sure where this paragraph goes, other than to highlight a different study by these authors? Are there regions identified in this region of the Arctic where understanding the processes would be particularly useful and how might that be achieved?

*>We agree – as we are not conducting isotope-based experiments in this research, we have removed this paragraph.*
*Line number in TC manuscript: 547*

L532/Section 4. This section starts off discussing areas of shallow mixed layer depth then merges into a discussion comparing DMSP:chl a ratios generally in the Arctic. What point(s) is being conveyed?

*>Here, we aim to characterize the general structure of our observations - to state that we observed elevated DMSP:chl in shallow MLD regions with a mixed assemblage, under potential light stress. We also want to compare these observations with others made in the region, for context.*

L552. Again, it is unclear what point is being made here - the range in DMSP:chl is > 3-fold, (52 - 182), why does the final sentence conclude low variability in DMSP:chl?

*>We have rephrased this section to better reflect that we cannot draw conclusions on the role of taxonomy in controlling DMSP:Chl a ratios - though the variability in DMSP:chl is relatively high, the variability in taxonomy appears low.*
*Line number in TC manuscript: 590*

L564. This section could be usefully focused. It would be more relevant to focus on what this dataset shows. The paragraph from L565 simply reviews previous studies and seems superfluous. As it stands, it is unclear what the authors conclude. The correlation between sea ice cover and DMS/P is negative but station-specific data suggests enhanced DMSP:chl ratios near the ice edge?

*>We have re-written this paragraph to focus the ideas presented. We start with a brief overview of some previous results examining potential ice effects on DMS/P cycling, and use this to provide a context for our work. We have significantly shortened the paragraph to focus on the most important messages.*
*Line number in TC manuscript: 605*

L558. Does the MIMS data pick-up ice edge effects on DMS concentration, i.e. is DMS related to salinity when passing through marginal ice zones or ice edges, over shorter distances than the whole dataset?

*>We have now clarified that the ice-edge effects on DMS were observed using MIMS.*

*Line number in TC manuscript: 636*

L605 Again this section reads very much like a review, with little focus on what this particular study demonstrates.

*>We agree and have removed this short section, as it does not speak to our specific results.*

Minor points

L144 - use CAA instead of Canadian. . .

*>Thank you, this has been corrected.*

L196 –should read 'convert DMSO to DMS'

*>Thank you, this has been corrected.*

L266, more correctly A is proportion of sea ice cover, rather than percentage.

*>Thank you, this has been corrected.*

Figure 1. GL - Greenland not GD; and please define CAA

*>Thank you, this has been corrected.*

Figure 2.please explain what the red dots are on the DMS graph, possibly station mesurements? This would strengthen the suggestion made on L595+

*>I have added the note that these are station measurements.*

Figure 3. please explain what the red dots are?

*>I have added the note that these are station measurements.*

Figure 6. Some more information in the legend would be useful, for instance, how are the data compiled, what exactly is illustrated? The total number of points for each dataset would also be useful.

*>We have updated this figure legend to reference the data sources and the total number of points.*

Dear Anonymous Referee #3,
Thank you for your thoughtful criticism of our work. We have reread and revised our manuscript according to the corrections you provided, including recalculating our wind data to a height of 10 meters. Please find responses to each of your points, below. We have done our best to address each statement carefully, and look forward to your responses. An updated manuscript is also available.

Best,
Tereza Jarnikova,
PhD student, UBC

Interactive comment on "The distribution of
methylated sulfur compounds, DMS and DMSP, in
Canadian Subarctic and Arctic marine waters
during summer, 2015" by Tereza Jarníková et al.
Anonymous Referee #3

1. General Comments In this paper, authors measured the concentrations of DMS
and DMSP in the surface seawater from the Labrador Sea to the Canadian Arctic
Archipelago during the summer of 2015. In addition to the distributions of DMS/P concentrations
with information of seawater parameters (hydrographic parameters, taxonomic
compositions, and sea ice cover) in this area, they detected the abrupt increases
in the DMS concentration at the front of hydrographic parameters by measurement with
fine spatial resolution. The data obtained from this observation contributes to the accumulation
of database in the Canadian Arctic waters and to discussion on the response
of DMS to changes occurring in the Arctic Ocean. This paper would be acceptable
if the authors reconsider and correct the part described in Specific Comments and

Technical Corrections.
2. Specific Comments
(1) Authors conclude that the results obtained from this cruise (shallow MLDs and
mixed phytoplankton assemblages) support the results of previous studies by Gabric
et al. (2004) and Levasseur (2013). I wonder whether the spatial changes such as
the differences between the sea-ice free area and the locations where sea ice has just
melted can be compared with the result caused by sea ice reduction in future Arctic
waters.

>*We do not believe that it is possible to compare recent sea-ice melt regions with a highly-stratified ice-free Arctic - it is likely that the processes governing DMS production in a recent ice-melt region (eg ice diatom water column release) are quite different from those in the stratified ocean. However, interestingly, we believe that our technique may be able to capture ice-melt effects.*

(2) The authors discuss the sharp increase in DMS concentration obtained from the
measurement of high spatial resolution. Then what happened where the DMS concentration
sharply decreased for example at 2600km, 3600km, 4000km, and 7600km?

>*As these are transect measurements, our paper suggests that the areas of sharp increase
correspond to encountering watermasses with high DMS production - we discuss the potential*

*role of fronts in DMS production later in the paper. The regions mentioned here are directly after the spikes in DMS - it may be inferred that the ship left the high-DMS zones at this time.*

(3) L262: In Equation (1), do you need to include the DMS concentration in the surface atmosphere? When you use this equation, you need to mention about omitting this.

*We do not include any surface atmosphere DMS in our calculation. We mention that we are assuming no surface DMS in the revised manuscript.*
*Line in track changes manuscript: 279*

(4) L269: The exchange coefficient of the flux calculation is a function of the wind speed at the 10m height from the sea surface. Please note that the height of the anemometer. And if the height is far from 10m, then you need to mention about the influence of the height difference.

*>This is an issue that has been brought to our attention. We have corrected the wind data to 10m above the surface and recalculated flux accordingly. The supporting data for the manuscript includes a figure showing the slight difference between the two windspeeds and fluxes.*
*Line in track changes manuscript: 289*

3. Technical Corrections
(1) When reading bg-2017-337.pdf, I found the following typing mistakes. I think that there are other typing mistakes in this draft, so reread and fix them.

(2) About the use of "Figure" and "Fig", if "Figure" is used at the beginning of a sentence and "Fig." is used in sentences, "Figure 3" at L317 is not at the beginning of the sentence, is this correct?

*>To minimize stylistic problems, I've written out the word "Figure" everywhere.*

(3) Figure 5 is referred (L329) prior to Figure 4 (L368). You need to swap Figure 4 and Figure 5.

> Thank you, this has been fixed.

(4) Table 4 is referred prior to Table 3. You need to swap Table 3 (L375) and Table 4

*>Thank you, this has been fixed.*

(L291). The order of Table 3 and Table 4 is also the same

*>Thank you, this has been fixed.*

(5) Is the writing style "km 7000" in L309 correct? This notation can be found in several other places such as L309, L322, L326, L331, L334, L336, L343, L346, L386, L403, L590, and L598.

*>This style is commonly used in geographic survey literature. We use it consistently in this paper as we believe it to be both concise and unambiguous.*

(6) In "Reference list", journal names written by full notation and abbreviations are mixed.

*>This has been corrected using the standard abbreviations.*

(7) In Figure 2, gray shaded areas denote not only the part of sharp increase in DMS concentration, but also the part of its high concentration (and its sharp decrease). If you want to highlight only the part of its sharp increase, carefully mark the part only where the DMS concentration increases.

*>Our approach was to highlight localized regions of DMS accumulation, which include both sharp increases and decreases.  We believe that showing the entire signal for each DMS pulse helps the reader visually compare coherence between DMS features and other hydrographic variables*

(8) List of technical errors L53: Is "three order" correct?

*>This is written as "can vary by three orders of magnitude", and a citation is provided.*

L59: At the end of L59, not period but comma.

*>        Thank you, this has been fixed.*
L70~L71: Invalid way to cite the references written in L70-L71; DMS emissions (Chang et al., 2011), (Mungall et al., 2016) ==> DMS emissions (Chang et al., 2011; Mungall et al., 2016)

*>        Thank you, this has been fixed.*

L92: mixed layer depth => mixed layer depth (MLD) L234, L280, L290, L534, L615,

*>        We have added the abbreviation MLD into instances of mixed layer depth mentions.*

L914: mixed layer depth => MLD

*>We believe it is correct to have both the full name and the abbreviation in this case, as it is a table caption.*

L128, L155, L200: The notation of "membrane inlet mass spectrometry (MIMS)" is to be used only at the first quotation, and the abbreviation (MIMS) should be used at subsequent citations.

*>Thank you, we have changed this.*

L130, L156 The notation of "OSSCAR" is same as my comment for "MIMS".

*We have changed this.*

L138: "(July 10-August 20, 2015)". Is parenthesis "( )" necessary?

*>We have adjusted this stylistically.*

L147: shallow, narrow straits => shallow and narrow straits

*>We would prefer to keep "shallow, narrow" straits.*

L263: Where => where

*>Thank you! Fixed.*

L306: 18nM => 18 nM

*>Thank you! fixed.*

L316, L317: 29nM => 29 nM, insert half-size space after "29" 52.31mmol => 52.31 nmol

*>Thank you! fixed.*

L314: Is "(measured with OSSCAR)" necessary? If so, ( ) is necessary?

*>Though this is implied by our instrumental setup, we've opted to remind the reader that DMSP is measured by OSSCAR here to emphasize the source of the data.*

L331: sea-air flux observed => sea-air flux calculated (or estimated)

*>We have changed this to 'calculated'.*

L348: (Fig 2c) => (Fig.2c) insert period after "Fig".

*>We have opted to write out "Figure" everywhere.*

L360: Is (G) in "Gradients (G) for each variable" necessary here?

*>We believe that this links the text to the formula shown, and have opted to keep it.*

L416: Quotation "Wolfe et al (2002)" at the end of L416 and the same author's quotation at the end of the sentence of L418 are duplicated (the latter is unnecessary).

*>Thank you! Fixed.*

L444: Fig 6a,=> Fig. 6a

*>We have opted to write out "Figure" everywhere.*

L446: Insert a half-size space between "(Fig. 6b)." and " The result„„".

*>Thank you! Fixed.*

L459: heavily => mainly

*>Thank you! Fixed.*

L461: drawn => obtained

*>Thank you! Fixed.*

L474: in the Northwest Subarctic => in the northwest subarctic

*>Thank you! Fixed.*

L474: Lizotte et al (2012) => Lizotte et al. (2012)

*>Thank you! Fixed.*

L500: Either "Previous" in L500 or "previously" in L501 is unnecessary.
*>Thank you! Fixed.*

L514: (Tremblay et al., 2011) in the end of sentence should be deleted.

*>Thank you! Fixed.*

L540: (Matrai et al. 1997) => (Matrai et al., 1997) Need comma after "et al."

*>Thank you! Fixed.*

L553, L554, L556: nM ug-1 => nmol ug-1

*>Thank you! Fixed.*

L560ïïjŽDMSP:Chl and HPLC => insert "a" after chl in italic. DMSP:Chl a and HPLC C4

*>Thank you! Fixed.*

L585: [30a] Reference?

*>Thank you! Fixed.*

L601-L602: Gali et al. (Gali et al., 2010) => Gali et al. (2010)

*>Thank you! Fixed.*

L613: Gabric et al (2005): Need period after "et al". Year of publish of this paper is 2004, not 2005.

*>Thank you! Fixed.*

L616: DMSP:Chl a ratios :Chl a should be italic.

*>Thank you! Fixed.*

L844: 20244. => 20244 (Delete period after 20244.)

*>Thank you! Fixed.*

L885: Adapted from (Coupel et al. 2015). => Adapted from Coupel et al. (2015).

*>Thank you! Fixed.*

L997: show denote =>Choose either "show" or "denote".

*>Thank you! Fixed.*

L1024-L1025: What are the grey area in these figures? Are the same as Fig.2?

*>Thank you! Fixed.*

L1040-L1041: Explain the panel (a) and (b).

*>Thank you! Fixed.*

[revised manuscript text omitted]